# Training-Free Test-Time Adaptation via Shape and Style Guidance for Vision-Language Models

**Shenglong Zhou[1], Manjiang Yin[2], Leiyu Sun[1], Shicai Yang[1], Di Xie[1]\*, Jiang Zhu[1]**

[1]Hikvision Research Institute
[2]University of Science and Technology of China

## Abstract

Test-time adaptation with pre-trained vision-language models shows impressive zero-shot classification abilities, and training-free methods further improve the performance without any optimization burden. However, existing training-free test-time adaptation methods typically rely on entropy criteria to select the visual features and update the visual caches, while ignoring the generalizable factors, such as shape-sensitive and style-insensitive factors. In this paper, we propose a novel shape and style guidance method (SSG) for training-free test-time adaptation in vision-language models, aiming to highlight the shape-sensitive (SHS) and style-insensitive (STI) factors. Specifically, SSG perturbs the raw test image with shape and style corruption operations, and measures the prediction difference between the raw and corrupted ones as perturbed prediction difference (PPD). Based on the PPD measurement, SSG reweights the high-confidence visual features and corresponding predictions, aiming to highlight the effect of SHS and STI factors during the test-time procedure. Furthermore, SSG takes both PPD and entropy into consideration to update the visual cache, aiming to maintain the stored sample with high entropy and generalizable factors. Extensive experimental results on out-of-distribution and cross-domain benchmark datasets demonstrate that our proposed SSG consistently outperforms previous state-of-the-art methods while also exhibiting promising computational efficiency.

## 1 Introduction

Vision-language models (VLMs) have demonstrated impressive effectiveness in downstream vision tasks, such as classification and generation. As the representative model, CLIP [1] is trained on extensive noisy image-text pairs to achieve multi-modality alignment, and leverages the learned embedding space to enable zero-shot image classification. Through the learned embedding space, images can be directly recognized by matching image features with text embeddings from different classes. Nevertheless, CLIP still faces challenges when encountering domain and distribution shifts during test-time inference. As a common real-world scenario, out-of-distribution (OOD) issues compromise CLIP's ability to maintain consistent cross-modal feature alignment, resulting in degraded performance.

To mitigate the out-of-distribution issues, test-time adaptation [2–6] is introduced to improve the model's generalization ability. Test-time adaptation only demands the unlabeled data stream to adapt the model during inference, which is beneficial for real-world utilization. Recently, along with the rapid development of large-scale vision-language models, studies about the test-time adaptation of vision-language models have been popular. As a pioneering work, Test-time Prompt Tuning [7] (TPT) is proposed to learn an adaptive prompt from the single test data, which is built on random

---

\*Corresponding Author

augmentations and self-entropy optimization. Following TPT, DiffTPT [8] brings in diffusion-based augmentations to provide more various views of the single data, thus facilitating better prompt tuning during test time. Although such prompt tuning methods show promising performance in terms of test-time adaptation, they require gradient descent to optimize learnable prompts which is time-consuming with large overhead, especially for large-scale vision-language models, and thus infeasible in some computationally constrained scenarios.

Different from prompt tuning, training-free methods utilize memory banks or caches to store the high-quality test samples and make use of these stored caches to improve the model's prediction adaptively. Taking TDA [9] as an example, it constructs a dynamic key-value cache with the pseudo labels of a stream of test samples as values and the corresponding extracted features as keys. During the test-time procedure, TDA exploits the valuable knowledge encoded in CLIP based on augmentations and entropy selection, while maintaining the cache update through historical entropy comparison. Then TDA produces output predictions based on the similarity between the test sample and stored data in dynamic caches. Compared with prompt tuning methods, training-free methods do not involve backpropagation optimization thus improving the inference efficiency obviously. However, existing training-free test-time adaptation methods typically focus on the entropy criteria, including using entropy to select the high-confidence samples from augmented views and updating the visual caches through entropy comparison, thus ignoring the exploration of generalizable factors which is important in the field of transfer learning.

In the field of transfer learning, generalizable factors, such as shape-sensitive factors and style-insensitive factors, are beneficial for models against domain shifts. Geirhos [10] conducts large-scale experiments and verifies the advantages of shape-sensitive factors to improve the model's robustness and transfer ability. MixStyle [11] proposes to force the model to learn style-insensitive factors by randomly mixing instance style statistics, which promotes the model's classification accuracy in domain generalization. DeYO [12] claims to combine the commonly positively-corelated with label factors with entropy to assist the backpropagation optimization of single-modality models such as ResNet18/ResNet50. However, vision-language models face the dual-modality misalignment, and generalizable factors have not been explored in the field of test-time adaptation in large-scale vision-language models such as CLIP, especially for the training-free methods.

In this paper, we propose shape and style guidance (SSG) method, which is the first exploration of generalizable factors for the training-free test-time adaptation in vision-language models. First of all, we demonstrate the benefits of generalizable factors from the perspective of disentangled theoretical analysis, and then design SSG to exploit shape-sensitive (SHS) and style-insensitive (STI) factors to represent generalizable factors. SSG proposes perturbation prediction difference (PPD) to determine SHS and STI factors quantitatively. Specifically, SSG perturbs the shape/style of test-time images efficiently, and measures the prediction difference between the raw image and the perturbed one as the PPD. Then, based on the PPD, SSG reweights the high-confidence visual features and corresponding predictions, aiming to highlight the effect of SHS and STI factors during the test-time procedure. Meanwhile, PPD can be utilized to cast as the cache criteria with entropy to update the dynamic visual cache, and the sample with lower entropy and higher PPD is stored in the visual cache, which means not only high confidence but also high SHS and STI factors are considered. We conduct comprehensive experiments on out-of-distribution and cross-domain benchmarks with two representative CLIP backbones ResNet-50 and ViT-B/16. Experimental results show SSG obviously outperforms state-of-the-art methods, while maintaining promising test-time inference efficiency.

## 2   Related Work

**Vision-Language Models.** Vision-Language Models have shown impressive generalization abilities through contrastive pre-training with numerous text-image pairs [1, 13–15]. CLIP [1] works by harmoniously aligning 400 million image-text pairs to predict the most relevant text description for a given image. Recently, adapting CLIP for downstream applications has garnered considerable attention and has been widely explored in various approaches [16–21]. Among these, CoOp [16] introduces the concept of learnable prompts [22–25], while CoCoOp [17] further refines these prompts by conditioning them on image embeddings to achieve better generalization. Maple [26], on the other hand, applies to prompt to both the vision and language branches, thereby enhancing the alignment between multimodal embeddings. These advancements have improved performance, yet they still require training on the target data with ground-truth labels. Differently, SSG aims to explore test-time adaptation, yet they still require training on the target data with ground-truth labels.

**Test-Time Adaptation.** Tent [2] minimizes generalization error on shifted data by employing test-time entropy minimization. Specifically for vision-language models, Test-time Prompt Tuning (TPT) [7] dynamically optimizes prompts during evaluation to improve zero-shot generalization. TPT works by generating multiple augmented versions of a test sample and minimizing the entropy of the model's predictions across these versions to ensure consistent results. Recent advancements like DiffTPT [8] leverage diffusion models to create semantically consistent augmented images for entropy minimization, while PromptAlign [27] aligns token statistics between test samples and the source distribution. TPS [28] adjusts per-class prototypes within the embedding space, and adapts by learning a shift vector for each prototype. DPE [29] proposes the dual-modality prototype learning for the test-time adaptation in vision-language models. However, these methods necessitate gradient descent operations on augmented images, posing significant computational and time challenges.

Recently, training-free test-time adaptation is popular which uses the cache model to facilitate the prediction of test samples in a non-parametric manner. Methods such as T3A [6] utilize prototypes as downstream classifiers, dynamically adjusting their weights to improve performance. For vision-language models, TDA [9] introduces both positive and negative caches to obtain high-quality test samples from the target domain. BoostAdapter [30] introduces the instance-aware boosting samples through the augmentation operations based on TDA. However, current training-free test-time adaptation methods in vision-language models typically ignore the generalizable factors such as shape-sensitive and style-insensitive factors, and we bring them into this field for the first time.

## 3 Method

### 3.1 Preliminaries

**Zero-Shot CLIP.** CLIP consists of two pre-trained parallel encoders, a visual encoder $E_v$ and a textual encoder $E_t$. For a $C$-class classification task, CLIP extracts the visual feature $f_{\text{test}}$ from a single test image $x_{\text{test}}$ and extracts textual features $\mathbf{W_C^T}$ from $C$ class-specific prompts. Then CLIP performs zero-shot predictions by computing the similarities between $f_{test}$ and $\mathbf{W_C^T}$ as

$$P\left(f_{\text{test}}\right) = f_{\text{test}}\mathbf{W_C^T}. \tag{1}$$

**Training-Free Test-Time Adaptation.** To avoid the heavy optimization burden while achieving promising performance, training-free test-time adaptation methods typically maintain a dynamic visual cache, which stores few-shot high-quality samples' visual features as keys $\mathbf{Q}$ and corresponding pseudo-labels as values $\hat{\mathbf{L}}$. Given the test-time visual feature $f_{\text{test}}$, the adapted predictions can be calculated by the stored cache as

$$P_{\text{cache}}\left(f_{\text{test}}\right) = A\left(f_{\text{test}}\mathbf{Q^T}\right)\hat{\mathbf{L}}, \tag{2}$$

where $A(z) = \alpha \exp(-\beta(1-z))$ is an adaptation function with a weighting factor $\alpha$ and a sharpness ratio $\beta$ in [18]. The final prediction is the combination of raw prediction $P\left(f_{\text{test}}\right)$ and the adapted prediction $P_{\text{cache}}\left(f_{\text{test}}\right)$.

**The Effect of Entropy.** As discussed in Sec.1, existing training-free test-time adaptation methods focus on entropy criteria to obtain the visual features and construct the visual cache, and we describe the effect of entropy in detail.

First, entropy involves the generation of visual features $f_{\text{test}}$. As presented in TPT [7], considering the test-time image $x_{\text{test}}$ is single, current methods typically augment the single image to various views as $\{x_{\text{test}}^1, ..., x_{\text{test}}^K\}$, then obtain the corresponding visual features $\{f_{\text{test}}^1, ..., f_{\text{test}}^K\}$ and prediction $\{P(f_{\text{test}}^1), ..., P(f_{\text{test}}^K)\}$. Then, entropy is utilized to select the high-confidence samples, which means features with lower entropy of prediction are selected. We term the selected visual features as $\mathbb{F}_e = \{f_{\text{test}}^{1*}, ..., f_{\text{test}}^{k*}\}$ and prediction as $\mathbb{P}_e = \{P(f_{\text{test}}^{1*}), ..., P(f_{\text{test}}^{k*})\}$. The test-time visual feature $f_{\text{test}}$ and corresponding prediction $P(f_{\text{test}})$ is the average of entropy-selected $\mathbb{F}_e$ and $\mathbb{P}_e$. Training-free test-time adaptation methods such as TDA [9] basically follow the above procedure to maintain promising performance. Though it is reasonable not to use augmented views in test-time adaptation, the performance can easily suffer by the complicated samples without various augmentation and entropy-based selection.

Second, entropy involves the update of the dynamic visual cache. Considering the visual cache's capacity is limited, entropy is selected as the cache criteria to determine which sample should be

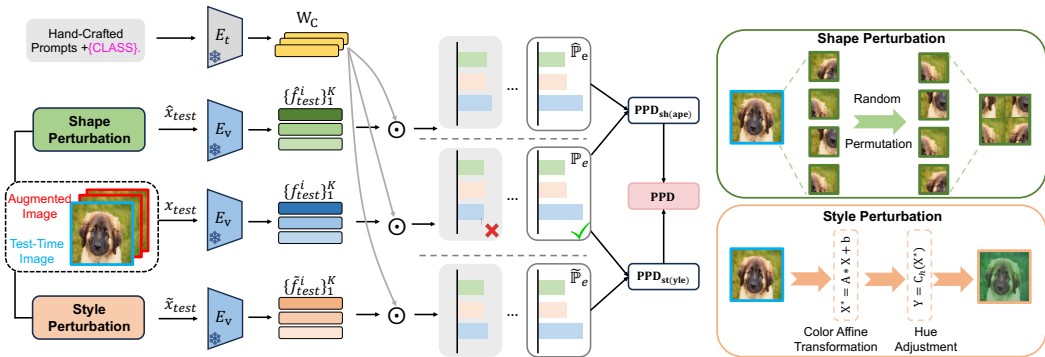

Figure 1: The generation of Permuted Prediction Difference (PPD). PPD consists of $\text{PPD}_{\text{st}}$ and $\text{PPD}_{\text{sh}}$, which is used to determine shape-sensitive and style-insensitive factors quantitatively. SSG proposes to perturb the shape of test-time images and measure the prediction difference as $\text{PPD}_{\text{sh}}$, where the perturbation is the patch-shuffle of image content. Meanwhile, SSG perturbs the style of test-time images and measures the prediction difference as $\text{PPD}_{\text{st}}$, where the perturbation is the colour affine transformation with hue adjustment.

stored. Specifically, the current test-time sample compares its entropy $\text{H}(f_{\text{test}}\mathbf{W}_{\mathbf{C}}^{\mathbf{T}})$ with stored samples' entropy $\text{H}(q_{\text{ent}}\mathbf{W}_{\mathbf{C}}^{\mathbf{T}})$ in the visual cache, where H denotes the entropy function and $q_{\text{ent}}$ is the 'uncertain' key in visual cache [9]. The lower entropy sample's visual feature and pseudo-label will be stored finally. Through the entropy comparison, the high-confidence sample is maintained in the visual cache.

Although entropy plays an important role in existing training-free test-time adaptation methods in VLMs, generalizable factors are totally ignored in the whole procedure, which can be proved beneficial as depicted below.

**Theoretical Analysis about Generalizable Factors.** As claimed in Sec.1, generalizable factors are beneficial for transfer learning. Following the previous works [12, 31], we give the theoretical analysis of the generalizable factors' important roles under the training-free test-time adaptation in vision-language models.

From the previous work [12], we can obtain that there is a disentangled latent vector $\mathbf{v}(x) = (\text{v}_1(x), \cdots, \text{v}_{d_v}(x))$ corresponding to the input image $x_{\text{test}}$ and feature $f_{\text{test}}$. For convenience, we assume $\mathbf{v}_i \in [0,1]$ and focus on binary classification, where label $\text{y} \in \{-1, 1\}$. Then we can define $\text{corr}_i^{\text{train}} = \text{corr}\left(\text{y}^{\text{train}}, \text{v}_i^{\text{train}}\right)$ is the correlation between the train label $\text{y}^{\text{train}}$ and the $i$-th factor $\text{v}_i^{\text{train}}$ corresponding to $x^{\text{train}}$, we can define $\text{corr}_i^{\text{test}} = \text{corr}\left(\text{y}^{\text{test}}, \text{v}_i^{\text{test}}\right)$ as the same way. After that, we can obtain different partitions from $\mathbf{v}$ based on above correlations:

$$\mathbf{v}_{pp} = \left\{\mathbf{v}_i \mid \text{corr}_i^{\text{train}} > 0, \text{corr}_i^{\text{test}} > 0\right\}, \mathbf{v}_{pn} = \left\{\mathbf{v}_i \mid \text{corr}_i^{\text{train}} > 0, \text{corr}_i^{\text{test}} \leq 0\right\}. \quad (3)$$

We regard $\mathbf{v}_{pp}$ as the generalizable factors which are commonly positively-correlated with the label, and $\mathbf{v}_{pn}$ as the spurious factors which are only positively-correlated with the label during the training-time. As a result, we have a theoretical conclusion that a sample $x \in \mathcal{X}^{\text{test}}$ is a harmful sample for training-free test-time adaptation in the vision-language model if it satisfies the following condition:

$$\begin{aligned}
\mathbf{v}_{pp}(x) \cdot \left(\mathbb{E}_{x' \sim \mathcal{X}_{+1}^{\text{test}}}\left[\mathbf{v}_{pp}\left(x'\right)\right] - \mathbb{E}_{x' \sim \mathcal{X}_{-1}^{\text{test}}}\left[\left[\mathbf{v}_{pp}\left(x'\right)\right]\right]\right) \\
+ \mathbf{v}_{pn}(x) \cdot \left(\mathbb{E}_{x' \sim \mathcal{X}_{+1}^{\text{test}}}\left[\mathbf{v}_{pn}\left(x'\right)\right] - \mathbb{E}_{x' \sim \mathcal{X}_{-1}^{\text{test}}}\left[\mathbf{v}_{pn}\left(x'\right)\right]\right) < 0,
\end{aligned} \quad (4)$$

where $\mathcal{X}_y^{\text{test}} = \{\mathbf{x} \mid (\mathbf{x}, \text{y}) \in \mathcal{D}^{\text{test}}, \text{y} = y\}$. Though we get the basically same conclusion as shown in [12], we give the insightful proof process, which is specially designed for training-free scenarios and is different from the case of entropy optimization. The detailed proof is provided in the Appendix.

From the Eq. 4, we can obtain that the sample with high confidence can be harmful, thus generalizable factors $\mathbf{v}_{pp}$ should be highlighted and spurious factors $\mathbf{v}_{pn}$ should be avoided, which approves the claims about the benefits of generalizable factors for training-free test-time adaptation in VLMs. Detailed analysis can be found in the Appendix.

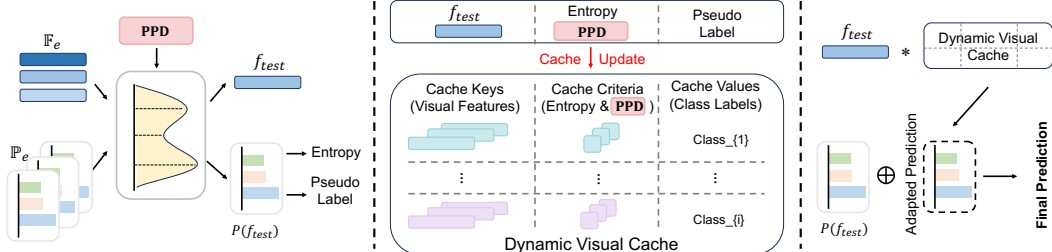

Figure 2: The effect of Permuted Prediction Difference (PPD). First, based on PPD, SSG reweights the entropy-selected augmented visual features $\mathbb{F}_e$ to get $f_{\text{test}}$ and $\mathbb{P}_e$ to get $P(f_{\text{test}})$, which highlights the shape-sensitive factors and style-insensitive factors. Second, PPD can be applied with entropy to cast as the cache criteria during the update of the visual cache, which means that not only high confidence is considered, but also high shape-sensitive and style-insensitive factors are considered. Finally, the reweighted visual feature $f_{\text{test}}$ queries the updated dynamic visual cache to obtain the adapted predictions, and generate the final predictions.

## 3.2 Shape and Style Guidance (SSG)

To highlight the generalizable factors, we leverage the prominent shape-sensitive [10] and style-insensitive factors [11] to represent them. In detail, we design shape and style guidance (SSG) to bring in the shape-sensitive and style-insensitive factors. First, SSG designs measurement methods to determine shape-sensitive and style-insensitive factors quantitatively. Then, SSG utilizes the designed measurement to affect the training-free test-time adaptation procedure in addition to entropy criteria. It is worth mentioning that we consider the test-time adaptation procedure with augmented images for the general case.

**Shape-Sensitive Factors Measurement.** To determine the shape-sensitive factors, SSG proposes to perturb the shape of test-time images first, and measure the prediction difference between the raw image and the perturbed one. The larger the difference is, the more the test-time image is affected by shape-sensitive factors. Specifically, given the augmented images $\{x_{\text{test}}^1, ..., x_{\text{test}}^K\}$, SSG utilizes the shape perturbation to obtain the corrupted images $\{\hat{x}_{\text{test}}^1, ..., \hat{x}_{\text{test}}^K\}$. By the way, considering the simplicity and effectiveness, SSG chooses the patch-shuffle operation to permute the image, which corrupts the shape of the object but preserves local information [12, 32]. We also utilize the entropy to select the high-confidence visual features $\hat{\mathbb{F}}_e$ and prediction as $\hat{\mathbb{P}}_e$. After that, based on the pseudo-label from raw prediction, we measure the difference between the raw prediction $\mathbb{P}_e$ and the perturbed one $\hat{\mathbb{P}}_e$, and term it as Shape Perturbed Prediction Difference ($\text{PPD}_{\text{sh}}$) as

$$\text{PPD}_{\text{sh}}\left(\mathbb{P}_e, \hat{\mathbb{P}}_e\right) = \left(\mathbb{P}_e - \hat{\mathbb{P}}_e\right)_{\mathbf{y}}, \tag{5}$$

where $\mathbf{y}$ represents the prediction of a specific class according to the pseudo-label from the raw prediction.

**Style-Insensitive Factors Measurement.** Besides the shape-sensitive factors, SSG also proposes to determine the style-insensitive factors. Following the same way, SSG perturbs the style of the test-time image, and measures the prediction difference. The smaller the difference is, the more the test-time image is affected by style-insensitive factors. Specifically, SSG utilizes the style perturbation to obtain the corrupted images $\{\tilde{x}_{\text{test}}^1, ..., \tilde{x}_{\text{test}}^K\}$. Though other advanced style translation operations such as instance statistic permutation or fourier-based style translation can be applied, SSG chooses the efficient colour transformation along with hue adjustment to perturb the style of the image, considering the efficiency of test-time adaptation. After the entropy selection and difference measurement in the same way, SSG obtains the Style Perturbed Prediction Difference ($\text{PPD}_{\text{st}}$)

$$\text{PPD}_{\text{st}}\left(\mathbb{P}_e, \tilde{\mathbb{P}}_e\right) = \left(\mathbb{P}_e - \tilde{\mathbb{P}}_e\right)_{\mathbf{y}}. \tag{6}$$

After obtaining the $\text{PPD}_{\text{sh}}$ and $\text{PPD}_{\text{st}}$ measurements, SSG combines them as Perturbed Prediction Difference (PPD) to influence the training-free test-time adaptation procedure. Because SSG aims to

highlight the shape-sensitive and style-insensitive factors, PPD is calculated as

$$\mathrm{PPD}\left(\mathbb{P}_e\right) = \mathrm{PPD}_{\mathrm{sh}}\left(\mathbb{P}_e, \hat{\mathbb{P}}_e\right) - \mathrm{PPD}_{\mathrm{st}}\left(\mathbb{P}_e, \tilde{\mathbb{P}}_e\right). \tag{7}$$

The whole and detailed process of obtaining PPD is shown in Fig. 1, along with the specific operations in $\mathrm{PPD}_{\mathrm{sh}}$ and $\mathrm{PPD}_{\mathrm{st}}$.

**The Effect of PPD.** As shown in Fig. 2, PPD can be deeply influenced the training-free test-time adaptation procedure in VLMs. First, SSG reweights the entropy-selected augmented visual features $\mathbb{F}_e$ by PPD to get $f_{\mathrm{test}}$, rather than the simple average operation as stated before, shown as

$$f_{\mathrm{test}} = \sum_{i=1*}^{k*} \mathrm{PPD}\left(\mathbb{P}_e\right)_i * f_{\mathrm{test}}^{i*}. \tag{8}$$

It is noticeable that PPD in Eq. 8 is processed by the softmax normalization to maintain the raw feature scale. Meanwhile, the same operation can be applied to entropy-selected augmented prediction $\mathbb{P}_e$. Through the above operations, SSG highlights the shape-sensitive and style-insensitive factors in high-confidence samples, which can further improve the model's generalization ability during the test time.

Second, PPD can be applied with entropy to cast as the cache criteria during the update of the visual cache, such as $\mathrm{H}(f_{\mathrm{test}}\mathbf{W}_{\mathbf{C}}^{\mathbf{T}}) - \mathrm{PPD}$. It means that not only high-confidence is considered, but also high shape-sensitive and style-insensitive factors are considered. Meanwhile, the stored visual feature also obtains the generalizable factors as shown in Eq. 8, which further boosts the test-time performance. Finally, the reweighted visual feature $f_{\mathrm{test}}$ queries the updated dynamic visual cache to obtain the adapted predictions, and generate the final predictions.

## 4 Experiments

### 4.1 Experimental Settings

**Datasets.** We follow previous works [7, 8] to evaluate our method on two benchmarking scenarios, including out-of-distribution benchmark and cross-domain benchmark. Out-of-distribution benchmark aims to evaluate the model's robustness to natural distribution shifts on 4 ImageNet [33] variants, including ImageNet-A [34], ImageNet-V2 [35], ImageNet-R [36], and ImageNet-Sketch [37]. The cross-domain benchmark aims to evaluate the transferring performance on 10 diverse recognition datasets, including FGVCAircraft [38], Caltech101 [39], StandfordCars [40], DTD [41], EuroSAT [42], Flowers102 [43], Food101 [44], OxfordPets [45], SUN397 [46], and UCF101 [47]. We follow the split in CoOp [16], and more details are shown in Appendix.

**Baselines.** We first compare SSG with the zero-shot CLIP prediction along with the ensemble version of 80 hand-crafted prompts from [1]. Then, we compare our method SSG with test-time adaptation approaches for CLIP. For the prompt tuning methods, we compare SSG with TPT [7], DiffTPT [8], and PromptAlign [48]. We also compare with recent prototype learning methods including TPS [28] and DPE [29]. For the training-free methods, we compare SSG with TDA [9], DMN-ZS [49] and BoostAdapter [30] with promising performance. Besides the test-time adaptation methods, we also show the performance of the representative work CoOp [16], Co-CoOp [17] and Maple [26].

**Implementation Details.** We adopt two types of backbone including ResNet-50 [50] and ViT-B/16 [51] as the visual encoder of CLIP. Following TPT [7] and TDA [9], we set the batch size as 1 and generate 63 augmented views for each test image, while setting the $k*$ as the top-10%. All experiments are conducted on a single 24GB NVIDIA RTX 4090 GPU. We build our SSG on the naive cache-based method, which only contains the positive cache and can be considered as the TDA [9] without a negative cache. For the cache implementation details, we follow the TDA [9] default setting for the positive cache. Specifically, the utilised cache in SSG is a dynamic key-value cache, whose memory size is 3 for all datasets. Considering our SSG is plug and play for existing training-free methods, so we also provide the enhanced version by combining BoostAdapter with SSG as the new version $\mathrm{SSG}^+$.

Table 1: Performance comparisons on out-of-distribution benchmark. We present top-1 accuracy(%) and employ both ResNet-50 and ViT-B/16 visual backbones of CLIP. The best results are highlighted in **bold**, and the second best results are highlighted in underline. The standard deviation is in brackets

| Method | ImageNet-A | ImageNet-V2 | ImageNet-R | ImageNet-S | OOD Average |
|---|---|---|---|---|---|
| CLIP-ResNet-50 [1] | 21.83 | 51.41 | 56.15 | 33.37 | 40.69 |
| Ensemble | 23.24 | 52.91 | 60.72 | 35.48 | 43.09 |
| CoOp [16] | 23.06 | 55.40 | 56.60 | 34.67 | 42.43 |
| TPT [7] | 26.67 | 54.70 | 59.11 | 35.09 | 43.89 |
| DiffTPT [8] | 31.06 | 55.80 | 58.80 | 37.10 | 45.69 |
| TDA [9] | 30.29 | 55.54 | 62.58 | 38.12 | 46.63 |
| DMN-ZS [49] | 28.57 | 56.12 | 61.44 | 39.84 | 46.49 |
| TPS [28] | 30.48 | 54.96 | 62.87 | 37.14 | 46.36 |
| DPE [29] | 30.15 | 56.72 | 63.72 | **40.03** | 47.66 |
| BoostAdapter [30] | 35.12 | 56.14 | 62.66 | 38.87 | 48.20 |
| **SSG (Ours)** | 31.54 (0.25) | 56.78 (0.18) | 63.77 (0.20) | 39.11 (0.12) | 47.78 (0.19) |
| **SSG$^+$ (Ours)** | **35.92** (0.22) | **57.29** (0.17) | **63.91** (0.22) | 39.51 (0.11) | **49.16** (0.18) |
| CLIP-ViT-B/16 [1] | 47.87 | 60.86 | 73.98 | 46.09 | 57.20 |
| Ensemble | 49.89 | 61.88 | 77.65 | 48.24 | 59.42 |
| CoOp [16] | 49.71 | 64.20 | 75.21 | 47.99 | 59.28 |
| Co-CoOp [17] | 50.63 | 64.07 | 76.18 | 48.75 | 59.91 |
| TPT [7] | 54.77 | 63.45 | 77.06 | 47.94 | 60.81 |
| DiffTPT [8] | 55.68 | 65.10 | 75.00 | 46.80 | 60.52 |
| PromptAlign [48] | 59.37 | 65.29 | 79.33 | 50.23 | 63.55 |
| TDA [9] | 60.11 | 64.67 | 80.24 | 50.54 | 63.89 |
| DMN-ZS [49] | 58.28 | 65.17 | 78.55 | **53.20** | 63.80 |
| TPS [28] | 60.08 | 64.73 | 80.27 | 49.95 | 63.76 |
| DPE [29] | 59.63 | 65.44 | 80.40 | 52.26 | 64.43 |
| BoostAdapter [30] | 64.53 | 65.51 | 80.95 | 51.28 | 65.57 |
| **SSG (Ours)** | 62.02 (0.14) | 65.32 (0.12) | 81.33 (0.18) | 52.13 (0.13) | 65.20 (0.13) |
| **SSG$^+$ (Ours)** | **65.16** (0.18) | **65.84** (0.14) | **81.54** (0.15) | 52.42 (0.13) | **66.24** (0.15) |

## 4.2 Experimental Results

**Out-of-distribution Benchmark.** As shown in Table 1, we compare our SSG with other state-of-the-art methods on 4 out-of-distribution ImageNet variants. Compared with test-time prompt tuning methods TPT, our SSG improves the average accuracy by about 3.9% on the ResNet-50 backbone and 4.4% on the ViT-B/16 backbone. Compared with prototype learning methods, both our SSG and SSG$^+$ show superior performance, especially for the ViT-B/16 backbone. It is worth mentioning that, our SSG does not demand the heavy optimization burden compared with prompt-tuning and prototype learning methods, while achieving better performance. Compared with the training-free test-time adaptation method TDA and DMN-ZS, our SSG provides an obvious improvement. Compared with the current SOTA method BoostAdapter, our SSG almost outperforms it on all datasets except for ImageNet-A. Meanwhile, our SSG$^+$ further enhances the performance compared with BoostAdapter, which verifies the effectiveness of our SSG.

**Efficiency Comparison.** As the training-free test-time adaptation methods, the inference efficiency is an important advantage against other methods such as test-time prompt learning and prototype learning methods. Therefore, we compare our SSG with other test-time adaptation methods for vision-language models. We evaluate different methods on the ImageNet-A dataset and the experiments are conducted on a single 24GB NVIDIA RTX 4090 GPU. In our SSG,

Table 2: Comparison results in terms of efficiency (testing time) and effectiveness (accuracy)

| Method | Testing Time | OOD Accuracy | Gain |
|---|---|---|---|
| CLIP [1] | ~3 min | 57.20 | - |
| TPT [7] | 2 h 36 min | 60.81 | +3.61 |
| DiffTPT [8] | >5 h | 60.52 | +3.32 |
| TDA [9] | ~15 min | 63.89 | +6.69 |
| DPE [29] | ~28 min | 64.43 | +7.23 |
| **SSG (Ours)** | ~15 min | **65.20** | **+8.00** |

the extra introduced computations are the shape and style perturbation, but these perturbation operations are extremely simple and do not bring in heavy computation overhead. Besides, the operations based on PPD such as reweighting or comparison are also lightweight. The detailed comparison results are shown in Table 2. Compared with test-time prompt tuning and prototype learning methods, it is obvious that our SSG shows a significant improvement in terms of testing time due to the design

Table 3: Performance comparisons on cross-datesets generalization. We also present top-1 accuracy(%) for all methods on two backbones of CLIP. The best results are highlighted in **bold**.

| Method | Aircraft | Caltech | Cars | DTD | EuroSAT | Flower | Food101 | Pets | SUN397 | UCF101 | Average |
|---|---|---|---|---|---|---|---|---|---|---|---|
| CLIP-RN-50 [1] | 15.66 | 85.88 | 55.70 | 40.37 | 23.69 | 61.75 | 73.97 | 83.57 | 58.80 | 58.84 | 55.82 |
| Ensemble | 16.11 | 87.26 | 55.89 | 40.37 | 25.79 | 62.77 | 74.82 | 82.97 | 60.85 | 59.48 | 56.63 |
| CoOp [16] | 15.12 | 86.53 | 55.32 | 37.29 | 26.20 | 61.55 | 75.59 | **87.00** | 58.15 | 59.05 | 56.18 |
| TPT [7] | 17.58 | 87.02 | 58.46 | 40.84 | 28.33 | 62.69 | 74.88 | 84.49 | 61.46 | 60.81 | 57.66 |
| DiffTPT [8] | 17.60 | 86.89 | 60.71 | 40.72 | 41.04 | 63.53 | **79.21** | 83.40 | 62.72 | 62.67 | 59.85 |
| TDA [9] | 17.61 | 89.70 | 57.78 | 43.74 | 42.11 | 68.74 | 77.75 | 86.18 | 62.53 | 64.18 | 61.03 |
| DPE [29] | 19.80 | **90.83** | 59.26 | 50.18 | 41.67 | 67.60 | 77.83 | 85.97 | 64.23 | 61.98 | 61.93 |
| BoostAdapter [30] | 18.93 | 88.48 | 59.67 | 43.85 | 44.40 | 68.25 | 78.78 | 85.75 | 62.83 | 64.42 | 61.54 |
| **SSG (Ours)** | 21.63 | 90.67 | 60.75 | 52.19 | 43.26 | 69.02 | 77.73 | 86.64 | 64.23 | 62.15 | 62.83 |
| **SSG$^+$ (Ours)** | **22.14** | 90.68 | **60.99** | **52.31** | **44.58** | **69.07** | 78.91 | 86.32 | **64.41** | **64.77** | **63.42** |
| CLIP-ViT-B/16 | 23.67 | 93.35 | 65.48 | 44.27 | 42.01 | 67.44 | 83.65 | 88.25 | 62.59 | 65.13 | 63.58 |
| Ensemble | 23.22 | 93.55 | 66.11 | 45.04 | 50.42 | 66.99 | 82.86 | 86.92 | 65.63 | 65.16 | 64.59 |
| CoOp [16] | 18.47 | 93.70 | 64.51 | 41.92 | 46.39 | 68.71 | 85.30 | 89.14 | 64.15 | 66.55 | 63.88 |
| CoCoOp [17] | 22.29 | 93.79 | 64.90 | 45.45 | 39.23 | 70.85 | 83.97 | 90.46 | 66.89 | 68.44 | 64.63 |
| TPT [7] | 24.78 | 94.16 | 66.87 | 47.75 | 42.44 | 68.98 | 84.67 | 87.79 | 65.50 | 68.04 | 65.10 |
| DiffTPT [8] | 25.60 | 92.49 | 67.01 | 47.00 | 43.13 | 70.10 | **87.23** | 88.22 | 65.74 | 62.67 | 65.47 |
| PromptAlign | 24.80 | 94.01 | 68.50 | 47.24 | 47.86 | 72.39 | 86.65 | 90.76 | 67.54 | 69.47 | 66.92 |
| TDA [9] | 23.91 | 94.24 | 67.28 | 47.40 | 58.00 | 71.42 | 86.14 | 88.63 | 67.62 | 70.66 | 67.53 |
| DPE [29] | 28.95 | 94.81 | 67.31 | 54.20 | 55.79 | **75.07** | 86.17 | 91.14 | 70.07 | 70.44 | 69.40 |
| BoostAdapter [30] | 27.45 | 94.77 | 69.30 | 45.69 | 61.22 | 71.66 | 87.17 | 89.51 | 68.09 | 71.93 | 68.68 |
| **SSG (Ours)** | 30.21 | 94.93 | 68.65 | **56.15** | 59.91 | 74.54 | 86.40 | 91.58 | 70.10 | 69.73 | 70.24 |
| **SSG$^+$ (Ours)** | **31.97** | **95.24** | **69.97** | 55.47 | **62.54** | 74.64 | 86.99 | **91.89** | **70.64** | **72.69** | **71.20** |

of the training-free method. Specifically, the computational efficiency of SSG is more than $8\times$ times higher than that of TPT and almost $2\times$ times higher than that of DPE, while improving the performance about 4.4% and 0.8%. Though SSG still gives a slower testing time compared with zero-shot CLIP, it obtains obvious OOD accuracy improvement of about 8.0%.

It is worth mentioning that, SSG applies the shape/style perturbation to the raw input images before the model forward procedure, and we concatenate them into a batch to feed into the model, thus the forward passes can be shared and maintain the forward time nearly unchanged. Meanwhile, due to the design of a triple number of inputs per sample to share the forward process for maintaining inference time, our SSG consumes more memory usage (3862 MiB) compared with TDA (3098 MiB). Our SSG increases the memory usage within an acceptable scope, as both 3862MiB and 3098MiB are generally considered to be acceptable memory usage ranges (< 4GB) in most practical scenarios. Thus, our SSG achieves a modest trade-off given the gains in accuracy and comparable inference time. Compared with DPE (9986 MiB), our SSG shows an obviously lower memory usage (3862 MiB) while achieving better OOD accuracy, which demonstrates the effectiveness of SSG.

**Cross-Domain Benchmark.** As shown in Table 3, we compare our SSG with other state-of-the-art methods on 10 fine-grained cross-domain datasets. Compared with the test-time prompt tuning methods TPT and DiffTPT, our SSG outperforms them obviously in the average accuracy. Compared with the prototype learning method DPE, our SSG gives a promising performance without any parameter optimization. Compared with the BoostAdapter method, our SSG also gives superior performance on 7 out of 10 datasets, showing the effectiveness of the whole design.

## 4.3 Ablation Studies

First, we conduct experiments to explore the influence of shape-sensitive and style-insensitive factors. Second, we conduct experiments to determine the influence of hyperparameters on the shape perturbation operation. Third, we conduct experiments to explore the effect of the PPD's operations including PPD reweighting and comparison.

**Shape-Sensitive and Style-Insensitive Factors.** SSG aims to bring in generalizable factors to assist test-time adaptation, thus designing $PPD_{sh}$ and $PPD_{st}$ to measure shape-sensitive and style-insensitive factors respectively. We conduct experiments to clarify the different effects of $PPD_{sh}$ and $PPD_{st}$, along with the full version as PPD shown in Eq. 7. There are three experiments, the first one only involves the $PPD_{st}$ measurement which means only the style-insensitive factors are brought

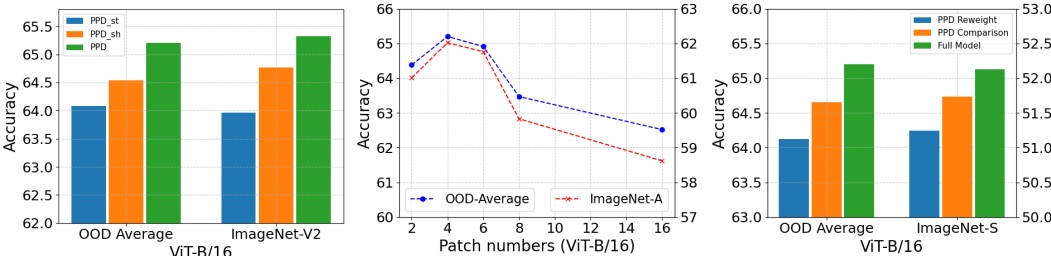

Figure 3: Ablation experiments about important components of SSG. Left: we conduct experiments to explore the influence of shape-sensitive and style-insensitive factors through $PPD_{sh}$, $PPD_{st}$ and PPD measurements. Middle: we conduct experiments to determine the influence of hyperparameters (patch number) on the shape perturbation operation. Right: we conduct experiments to explore the effect of the PPD's operations including PPD reweighting and comparison.

into the test-time adaptation procedure. The second one only considers the shape-sensitive factors and $PPD_{sh}$ measurement, and the last one is the full model consisting of both $PPD_{sh}$ and $PPD_{st}$, which is termed PPD.

We conduct experiments on the ViT-B/16 backbone, and the results on all OOD datasets and the single ImageNet-V2 are shown in Fig. 3 left. The results show that $PPD_{sh}$ achieves more achievement than $PPD_{st}$, which shows that shape-sensitive factors represent better generalizable factors than style-insensitive factors to some degree. Though style and shape are considered the disentangled factors in some cases, shape-sensitive factors are more straightforward for the generalizable concept. But it is worth mentioning that, the full model which utilizes the PPD shows the best performance, which means the style-insensitive factors are a good complement to shape-sensitive factors.

**Hyperparameters in Shape Perturbation Operation.** To obtain the $PPD_{sh}$ measurement, SSG introduces the shape perturbation operation to generate the shape-corrupted images. Specifically, $PPD_{sh}$ chooses the simple patch shuffle operation as the shape perturbation, and achieves the promising performance. As the important hyperparameter in patch shuffle, the patch number in the image is desired to be explored. We choose different patch numbers to conduct experiments on ViT-B/16, and show the results in Fig. 3 middle. The choice patch numbers range from 2 to 16, which covers the extreme cases in the regular patch shuffle operation.

According to the accuracy of ImageNet-R and ImageNet-A, we can find the patch number as {4, 6} performs the better performance, both smaller and larger patch numbers decrease the accuracy obviously, especially for the larger patch number such as 16. We analyze the results from the property of shape corruption. A small patch number such as 2 may not corrupt the semantic instances due to the small instance size, so the shape-sensitive factors are not captured well. For the large patch number of 16, the local information is totally corrupted, thus the prediction is easily affected by many factors rather than the shape-sensitive factors. Therefore, a proper patch number is demanded when applying the patch shuffle, and we choose patch number 4 in default for all main experiments, and the performance shows a stable improvement.

**Hyperparameters about the combination between** $PPD_{sh}$ **and** $PPD_{st}$. For the combination of $PPD_{sh}$ and $PPD_{st}$ in Eq. 7, considering the training-free methods' efficiency and no extra learnable parameters, we combine them in a simple way by the hyperparameter as 1 and achieve the best performance compared with alone $PPD_{sh}$/$PPD_{st}$ as shown in Fig. 3. We conduct experiments about different ratios to combine $PPD_{sh}$ and $PPD_{st}$, such as rescaling $PPD_{st}$ with 0.5/1.5/2.0 factors to obtain PPD. The OOD accuracy of SSG varies from 65.20 to 65.11/65.07/65.15. This result shows the performance range is relatively stable, which shows the robustness of SSG performance. Considering the situation in actual scenarios, the scale of $PPD_{st}$ can be simply set to 1.0.

**The Effect of PPD Operation.** Once obtained the PPD measurement, SSG applies it to reweight the high-confidence samples to generate the test-time visual feature, while assisting the visual cache selection along with the entropy criteria. To clarify the different effects of these two operations, we design three ablation experiments. The first one only involves the PPD reweighting, and the second one only involves the PPD comparison. The third one is the full model with both PPD reweighting and comparison. We conduct experiments on ViT-B/16, and show the accuracy results on all OOD datasets and the single ImageNet-S dataset in Fig. 3 right. According to the results,

Table 4: Comparison results based on the OpenCLIP (ViT-L/14) backbone.

| Method | ImageNet-A | ImageNet-V | ImageNet-R | ImageNet-S | OOD Average |
|--------|-----------|-----------|-----------|-----------|-------------|
| OpenCLIP [52] | 53.88 | 67.69 | 87.42 | 63.18 | 68.13 |
| TDA [9] | 61.27 | 68.42 | 88.41 | 64.67 | 70.69 |
| DPE [29] | 61.09 | **70.83** | 89.18 | 66.33 | 71.86 |
| **SSG (Ours)** | **62.85** | 69.97 | **89.67** | **66.56** | **72.26** |

we can find reweighting operation achieves better performance and a combination of them gets the best performance. It is worth mentioning that PPD comparison for the update of the visual cache is helpful for the PPD reweighting. It illustrates that even for the single test-time image without any augmentation, the PPD comparison still contributes to the construct of the visual cache.

**SSG on other Vision-Language models.** Our SSG can easily be applied to other vision-language models, and we follow DPE to involve the OpenCLIP (ViT-L/14) [52] as an example for a fair and public comparison. We show the comparison results in Table 4. We can observe that our SSG still outperforms training-free TDA by 1.57% on average across 4 datasets, showing that our method generalizes well to the larger-scale vision-language model.

## 5 Conclusion

In this work, we propose a novel shape and style guidance (SSG) method for the training-free test-time adaptation fields in VLMs. SSG first brings the generalizable factors such as shape-sensitive and style-insensitive factors into the test-time adaptation procedure. To determine the generalizable factors, SSG designs the shape perturbed prediction difference ($\text{PPD}_{\text{sh}}$) and style perturbed prediction difference ($\text{PPD}_{\text{st}}$) measurement, by utilizing the shape perturbation and style perturbation to corrupt the raw image. Based on the perturbed prediction difference (PPD), SSG reweights the high-confidence visual features for a better generalizable representation, and introduces PPD as the cache criteria along with the entropy.

Despite the promising performance of our SSG, it also has some limitations. Though SSG proposes to explore the generalizable factors, the current utilized shape and style perturbation operation is straightforward, and more advanced and efficient operations, such as fourier-based or channel-based methods, can be discussed in the future. At the same time, there are always some tasks (e.g., texture-based medical image analysis) that depend on style information, and how to explore the generalizable factors on these tasks is one promising research direction.

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
