# OpenReview forum: "Training-Free Test-Time Adaptation via Shape and Style Guidance for Vision-Language Models"
_NeurIPS.cc/2025/Conference — NeurIPS 2025 poster_

### Official Review · Reviewer_rBMJ · 2025-06-19

**Clarity:** 2
**Significance:** 2
**Originality:** 2
**Rating:** 4
**Confidence:** 4

**Summary:**

The paper proposes a method named Shape and Style Guidance (SSG) for training-free test-time adaptation in vision-language models.
It introduces the concepts of perturbed prediction difference (PPD) to measure the importance of shape-sensitive (SHS) and style-insensitive (SIS) factors during test-time adaptation.
SSG incorporates PPD into the test-time procedure by reweighting high-confidence visual features and corresponding predictions based on PPD.
The method updates the dynamic visual cache not only by entropy but also by considering PPD, allowing the storage of samples with both high confidence and generalizable factors.
The paper conducts a number of experiments and claims state-of-the-art performance under specific experimental settings.

**Questions:**

1. Novelty and Distinction from DeYO

 * Concern : The paper's novelty appears limited, as many core ideas and mechanisms closely resemble those in the DeYO method.

(1)PPD vs. PLPD: The proposed PPD (Perturbed Prediction Difference) and DeYO's PLPD (Pseudo-Label Probability Difference) seem to be conceptually identical. Both measure prediction differences to reveal sample dependencies on specific factors.

(2) Generalizable Factors vs. Commonly CPR (Commonly Positively-coRrelated) factor: The paper claims to introduce "generalizable factors" (e.g., shape and style), but these concepts are already discussed in DeYO, which emphasizes the importance of shape as a CPR factor. The addition of style seems to be a minor extension.

(3) Cache Update Strategy: The cache update strategy based on "lower entropy and higher PPD" appears to be very similar to DeYO's sample selection strategy based on "lower entropy and higher PLPD" (as shown in DeYO's Fig. 4).

(4) Theoretical Analysis: The section on "Theoretical Analysis about Generalizable Factors" seems highly similar to DeYO's section on "Entropy is Not Enough." Please clarify the unique theoretical contributions of this paper compared to DeYO. What new insights or theoretical advancements are provided that were not covered in DeYO?

* Request: Please provide a detailed comparison and clarification of how the proposed method fundamentally differs from DeYO. Highlight the specific innovations and contributions that go beyond what DeYO has already achieved.

2. Experimental Contributions

* Concern: The Shape Guidance proposed in this paper has already been introduced in DeYO. The addition of Style Guidance appears to be a minor alteration. If the authors can show that the inclusion of style significantly enhances the performance or robustness of the model in ways that shape alone cannot, this would to some extent address concerns about limited novelty.

* Request: Please provide experimental evidence or analysis that demonstrates the unique benefits of incorporating style in addition to shape. How does the inclusion of style guidance improve the model's performance or robustness in scenarios where shape guidance alone is insufficient?

**Ethical Concerns:**

["NO or VERY MINOR ethics concerns only"]

**Final Justification:**

I appreciate the authors for their thorough and well-organized rebuttal, which has addressed part of my concerns. While I feel the novelty is moderate, the additional clarifications and ablation studies offers an enhancement. Therefore, I raise my score to 4.

**Limitations:**

The paper mentions the computational overhead of previous methods like TPT but does not demonstrate a significant improvement in computational efficiency compared to other training-free methods.The authors should provide a detailed comparison of computational efficiency with other training-free methods, highlighting any specific optimizations or improvements that their method offers.

This work utilizes both shape perturbation and style perturbation to corrupt the raw image. However, it is reasonable not to heavily use augmented views in test-time adaptation, as such operations (e.g., cropping, color perturbations) may lead to the loss or displacement of key semantic information. Forcing consistency across these augmented views can suppress the model's ability to discern fine-grained semantics, thereby affecting the final prediction.

**Paper Formatting Concerns:**

No major formatting issues were observed.

**Quality:**

2

**Strengths And Weaknesses:**

Strengths:

Quality: The paper is generally easy to follow and the methodology is clearly described. The experiments cover various out-of-distribution and cross-domain benchmarks, and the results are presented in a structured manner.

Weaknesses:

Clarity: The paper fails to clearly articulate its differences from DeYO. The distinctions between the proposed method and DeYO are not well-explained, which affects the clarity of the paper.
(DeYO: Entropy is not enough for test-time adaptation: From the perspective of disentangled factors.arXiv preprint arXiv:2403.07366, 2024.)

Originality: The paper's novelty is questionable due to significant similarities with the DeYO method. For detailed concerns regarding originality, please refer to the question section below.

Significance: The improvements over previous methods are incremental. The overall significance of the contributions is somewhat limited by the lack of clear differentiation from existing work.

---

> ### Author Rebuttal · Authors · 2025-07-31
>
> Thanks for your comments, and we provide our feedback on questions (Q) and limitations (L) as follows.
>
> **Q1**: *"Novelty and Distinction from DeYO."*
>
> **A1**: Thanks for your feedback. We will provide a detailed comparison and clarification between our SSG and DeYO in the following points.
>
> First, corresponding to (4), we give **the theoretical analysis about generalizable factors for the training-free VLM TTA methods for the first time**. We appreciate the insight and theoretical contribution from DeYO, and we formally mention DeYO in Line151 and Line154, but DeYO only considers the entropy minimization back-propagation scenarios rather than considering the training-free (cache-based) cases. We extend the theoretical analysis to the training-free methods, and we think it is not trivial to accomplish this extension. Specially, DeYO calculates the gradient $\partial \operatorname{Ent}_\theta(\mathbf{x})/{\partial w}$ when adapting through entropy minimization loss to determine the update of $w$, then determines the change in the gap between the mean logits of samples belonging to different classes. However, our SSG utilises the properties of the cache-based method, which can directly determine the update of $w$ by the stored visual feature $\mathbf{v}(x)$ without the derivations from the gradient of entropy, such as $\Delta\left(w \cdot \mathbf{v}\left(\mathbf{x}^{\text {test}}\right)\right)=\Delta w(\mathbf{x})\cdot \mathbf{v}\left(\mathbf{x}^{\text {test}}\right) \approx \mathbf{v}(\mathbf{x}) \cdot \mathbf{v}\left(\mathbf{x}^{\text {test}}\right)$, where stored visual feature can be regarded as the residual update of the raw classifier. Then we can obtain the change in the gap between the mean logits of samples belonging to the two classes as Equation 7 in the Appendix. Though the conclusion is the same, the core derivation of the two methods is different, as shown in Line15-Line21 in the Appendix, and we reveal the theoretical support for the cache-based (training-free) methods in a reasonable way, which can not be intuitively supported by DeYO theory without any modification.
>
> Second, corresponding to (3), **our SSG is designed for the training-free VLM TTA settings, which are different from the DeYO settings.** DeYO aims to filter samples based on PLPD to benefit the entropy back-propagation, however, our SSG does not demand the entropy back-propagation. Our SSG aims to use PPD to highlight the augmented views in a novel reweight way, and supplement generalizable factors for the cache update. Based on the theoretical analysis, it is natural to highlight "lower entropy and higher PPD", but in terms of setting and specific operations, our SSG is different from DeYO. Moreover, DeYO only conducts the experiments on the single-modality scenarios, while our SSG conducts and verifies our design on large-scale vision-language models (CLIP) with promising performance and efficiency.
>
> Third, corresponding to (2), though SSG (PPD) and DeYO (PLPD) introduce the generalizable factors, **our SSG not only considers the shape-sensitive factors but also involves the style-insensitive factors**. Style-related factors are important for the transfer learning methods, which have been proven in MixStyle[1], FourierDG[2], DSU[3] and more works. Therefore, the introduction of style-insensitive factors is important for the generalizable factors, which makes it more comprehensive and sound. Meanwhile, we have conducted the ablation results about style-insensitive factors in Figure 3 left in the manuscript, and style-insensitive factors show a promising improvement, which is a good supplement to their importance. Furthermore, we supplement more ablation results in the following **A2**, and get the same conclusion.
>
> Corresponding to (1), we acknowledge that PPD measures prediction differences to reveal sample dependencies on specific factors, which is similar to DeYO's formula. Meanwhile, we also think it is a common and efficient operation which has been deployed in many different transfer learning fields like [4] [5], and it is reasonable to use this operation to determine the generalizable factors.
>
> Based on the above main points, we think our SSG is different from DeYO, and we will give more discussions about DeYO to make it more explicitly credited in Sections 3.1 and 3.2 in the final manuscript.
>
> [1] Domain generalization with mixstyle, 2021ICLR; [2] A fourier-based framework for domain generalization,2021CVPR; [3] Uncertainty modeling for out-of-distribution generalization, 2022ICLR; [4] Style Normalization and Restitution for Generalizable Person Re-identification, 2020CVPR; [5] Generalized lightness adaptation with channel selective normalization, 2023ICCV.
>
> **Q2**: *"Experimental Contributions. If the authors can show that the inclusion of style significantly enhances the performance or robustness of the model in ways that shape alone cannot, this would to some extent address concerns about limited novelty."*
>
> **A2**: Thanks for your suggestion. We have conducted detailed ablation results about the effect of PPD_st (style-insensitive factors) in Figure 3 left in the manuscript, and we supplement the average OOD accuracy based on CLIP-RN50 for further support. In the below table, the baseline method does not contain style-insensitive and shape-sensitive factors, PPD_st refers to style-insensitive factors, PPD_sh refers to shape-sensitive factors, and PPD refer the full model.
>
> | OOD Accuracy        | Baseline | PPD_st | PPD_sh | PPD    |
> | ------------------- | -------- | ------ | ------ | ------ |
> | SSG (CLIP-RN50)     | 46.33%   | 46.85% | 47.11% | 47.78% |
> | SSG (CLIP-ViT-B/16) | 63.46%   | 64.08% | 64.54% | 65.20% |
>
> From the above table, it is obvious that incorporating only the style-insensitive factors (PPD_st), the performance is improved from 46.33%/63.46% to 46.85%/64.08%, and further incorporation with shape-sensitive factors, the performance further enhance. The improved performance obviously demonstrates the effectiveness of the inclusion of style, and makes the generalizable factors more comprehensive and sound.
>
> **L1**: *"The authors should provide a detailed comparison of computational efficiency with other training-free methods, highlighting any specific optimisations or improvements that their method offers."*
>
> **A3**: Thanks for your suggestions.
>
> First, SSG utilised shape and style augmentation is extremely simple to implement. For the shape augmentation, we use the patch-shuffle operation, which can be efficiently implemented by `erarrange func` in `einops packages`. For the style augmentation, we use the colour transformation with hue adjustment, which can be easily implemented by the `kornia package`.
>
> Second, we apply the shape/style augmentation to the raw input images before the model forward procedure, and we concatenate them into a batch to feed into the model, which can be considered as the processing time is nearly similar.
>
> Third, after model feedforward and obtaining the logits, the following PPD calculation, reweighting and cache update is a simple mathematical calculation, which does not introduce much more inference burden.
>
> In conclusion, our SSG shows the promising inference efficiency, and we give the detailed inference time comparison below.
>
> | Method     | Data Process | Model Forward | Rest Part | Total  |
> | ---------- | ------------ | ------------- | --------- | ------ |
> | TDA        | 15 ms        | 82 ms         | 34 ms     | 132 ms |
> | SSG (Ours) | 21 ms        | 83 ms         | 35 ms     | 139 ms |
>
> It is obvious that SSG needs more data processing time due to the shape/style augmentation, and the other two parts need similar inference time, but the increased inference time is relatively small. Furthermore, we supplement the efficiency comparison in ImageNet-V, which contains 10000 images, in the following table.
>
> | Method     | Testing Time | OOD Accuracy | Gain  |
> | ---------- | ------------ | ------------ | ----- |
> | TDA        | 22 min       | 63.89        | +6.69 |
> | DPE        | 41 min  | 64.43        | +7.23 |
> | SSG (Ours) | 23 min 10 s`      | 65.20        | +8.00 |
>
> From the above table, even for the 1w image scale, our SSG only increase 1 minute compared with TDA while improving the OOD accuracy by about 1.3%. At the same time, our SSG only demands 56% inference time compared with DEP while achieving better performance.
>
> **L2**: *"It is reasonable not to heavily use augmented views in test-time adaptation, as such operations (e.g., cropping, colour perturbations) may lead to the loss or displacement of key semantic information. Forcing consistency across these augmented views can suppress the model's ability to discern fine-grained semantics, thereby affecting the final prediction."*
>
> **A4**: Thanks for your comments. We do not force consistency across the raw images and their shape or style augmented images. Actually, we use shape and style augmentation to corrupt the raw images' shape and style information, aiming to measure the shape-sensitive and style-insensitive factors to further represent the generalizable factors.

---

### Official Review · Reviewer_B6DJ · 2025-06-23

**Clarity:** 4
**Significance:** 3
**Originality:** 2
**Rating:** 5
**Confidence:** 3

**Summary:**

The paper presents a training-free test-time adaptation method using dynamic key-value cache with pseudo-labels. The key contribution is employing perturbed prediction difference (PPD) as weighting criteria for entropy-based cache updates. PPD computation relies on two generalizable factors: shape-sensitive (SHS) and style-insensitive (SIS) perturbations. The method demonstrates improvements over existing approaches across two benchmarks using different backbone architectures.

**Questions:**

1. How does PPD and the theorical analysis differ from DeYO's work, and shouldn't this prior work be more explicitly credited rather than claimed as a contribution?

2. Can you provide comparisons against simple baseline augmentations (e.g., random combinations of Gaussian noise and rotations) with PPD computation to demonstrate that your specific perturbation choices are essential? Additionally, please clarify whether the entropy computation uses the same shape and style perturbations as PPD, or different augmentations such as those employed in TPT?

3. Can you provide the missing ablation studies for ResNet-50, particularly showing the individual effects of PPD_sh and PPD_st, as well as the impact of patch shuffling on CNN-based architectures?

I am open to raising my evaluation score if the authors provide convincing responses to the questions outlined above and address the clarification points regarding target distribution access and the foundational method underlying SSG.

**Ethical Concerns:**

["NO or VERY MINOR ethics concerns only"]

**Final Justification:**

I appreciate the authors' responses to all of my questions. Their explanations have adequately resolved my initial concerns, and I will therefore raise my score to accept

**Limitations:**

yes

**Paper Formatting Concerns:**

No formatting issues

**Quality:**

3

**Strengths And Weaknesses:**

## **Strengths**
**Writing Quality:**  The paper is exceptionally well-written with clear, accessible sections and comprehensive explanations.

**Literature Positioning:** Authors effectively position their work as the first training-free test-time adaptation method to extend beyond entropy-only updates for visual-cache memory, incorporating style and shape factors.

**Method Design:**  The approach is elegant and intuitive. Using PPD as an entropy weighting factor represents an interesting innovation that yields good performance improvements.

**Computational Efficiency:** The method maintains low testing time with negligible computational overhead from additional shape and style augmentations.

## **Weaknesses**

**Theoretical Attribution:** The authors claim as a contribution the demonstration of generalizable factors through disentangled theoretical analysis (L68-70). However, this analysis directly replicates the theoretical framework from DeYO [1] without proper attribution. While the paper mentions following previous works (L151), it fails to clearly acknowledge that the theoretical analysis is identical to DeYO's contribution, which undermines the claimed novelty.

**Insufficient Baselines:** The paper lacks comparison against straightforward augmentation baselines that combine geometric and photometric transformations. Although Table 1 in supplementary materials shows some shape perturbations, this evaluation is insufficient to demonstrate that the specific choice of style and shape perturbations is crucial rather than simply computing PPD on any reasonable augmentation strategy.

**Limited Technical Novelty:** The core technical contribution, PPD, has been previously studied in DeYO under the name Pseudo-Label Probability Difference (PLPD). The authors do not clearly acknowledge that or distinguish their approach sufficiently from PLPD in DeYO. This raises questions about the technical novelty of the proposed method.

**Incomplete Ablations:** The left and middle ablations of Figure 3 are only shown for Vit-B/16 but not for ResNet-50. Given that CNNs process information differently than transformers, it would be valuable to assess how PPD_sh and PPD_st perform individually on CNN architectures, as well as understand the effect of patch shuffling.

### **Notes**

**Typo:** in L13, it should be SIS instead of STI

**Target Distribution Access:** In L91, the authors claim SSG lacks information about the target distribution. However, since SSG applies augmentations to test images and computes both entropy and PPD on them, it does have access to target distribution information. The distinction should be clarified between having access to target distribution versus training on it, as training-free methods can still operate on and extract information from test data.

**Method Foundation:** The experiments section mentions SSG+ as a combination of BoostAdapter and SSG, but it's unclear what existing method SSG builds upon. Given the similarity to TPT (entropy-based approach with SSG adding PPD weighting), clarification is needed about whether SSG extends TPT or represents an independent development of similar concepts.

[1] : Lee, Jonghyun, et al. "Entropy is not Enough for Test-Time Adaptation: From the Perspective of Disentangled Factors." ICLR 2024

---

> ### Author Rebuttal · Authors · 2025-07-31
>
> We sincerely thank you for your constructive comments, and we  provide our feedback on weaknesses (W) and questions (Q) as follows.
>
> **W1&W2&Q1**: *"How does PPD and the theoretical analysis differ from DeYO's work, and shouldn't this prior work be more explicitly credited rather than claimed as a contribution?"*
>
> **A1**: Thanks for your feedback. We will depict the difference between our SSG and DeYO in the following points.
>
> First, we give the **theoretical analysis about** **generalizable factors for the training-free VLM TTA methods for the first time**. We appreciate the insight and theoretical contribution from DeYO, and we formally mention DeYO in Line151 and Line154, but DeYO only considers the entropy minimization back-propagation scenarios rather than considering the training-free (cache-based) cases. We extend the theoretical analysis to the training-free methods, and we think it is not trivial to accomplish this extension. Specially, DeYO calculates the gradient $\partial \operatorname{Ent}_\theta(\mathbf{x})/{\partial w}$ when adapting through entropy minimization loss to determine the update of $w$, then determines the change in the gap between the mean logits of samples belonging to different classes. However, our SSG utilises the properties of the cache-based method, which can directly determine the update of $w$ by the stored visual feature $\mathbf{v}(x)$ without the derivations from the gradient of entropy, such as $\Delta\left(w \cdot \mathbf{v}\left(\mathbf{x}^{\text {test}}\right)\right)=\Delta w(\mathbf{x})\cdot \mathbf{v}\left(\mathbf{x}^{\text {test}}\right) \approx \mathbf{v}(\mathbf{x}) \cdot \mathbf{v}\left(\mathbf{x}^{\text {test}}\right)$, where stored visual feature can be regarded as the residual update of the raw classifier. Then we can obtain the change in the gap between the mean logits of samples from two classes as Equation 7 in the Appendix. Though the conclusion is the same, the core derivation of the two methods is different, as shown in Line15-Line21 in the Appendix and stated in Line 165 in the manuscript. We reveal the theoretical support for the cache-based (training-free) methods in a reasonable way, which can not be intuitively supported by DeYO theory without any modification.
>
> Second, **our SSG is designed for the training-free VLM TTA settings, which are different from the DeYO settings.** DeYO aims to filter samples based on PLPD to benefit the entropy back-propagation. However, our SSG does not demand the entropy back-propagation, and aims to use PPD to highlight the augmented views in a reweight way, and supplement generalizable factors for the cache update. In terms of setting and specific operations, our SSG is different from DeYO. Moreover, DeYO only conducts the experiments on the single-modality scenarios, while our SSG conducts and verifies our design on large-scale vision-language models (CLIP) with promising performance and efficiency.
>
> Third, though SSG (PPD) and DeYO (PLPD) introduce the generalizable factors, **our SSG not only considers the shape-sensitive factors but also involves the style-insensitive factors** motivated by the MixStyle, which makes the generalizable factors more comprehensive and sound. And the ablation results about style-insensitive factors in Figure 3 left in the manuscript show the promising improvement, which is a good supplement to style-insensitive factors, as denoted in **A5**. As for the specific operations of PPD, we think it is common to measure prediction differences to reveal specific factors.
>
> Based on the above main points, we think our SSG is different from DeYO, and we will give more discussions about DeYO to make it more explicitly credited in Sections 3.1 and 3.2 in the final manuscript.
>
> **W2-1**: *"The paper lacks comparison against straightforward augmentation baselines that combine geometric and photometric transformations."*
>
> **A2**: Thanks for your valuable comments. We supplement the comparison experiments between: (a) straightforward augmentation baselines, which means our SSG model without PPD-related operations but considers geometric and photometric transformations in the augmentations, and (b) our full SSG model.
>
> | OOD Accuracy    | (a)    | (b)    | OOD Accuracy        | (a)    | (b)    |
> | --------------- | ------ | ------ | ------------------- | ------ | ------ |
> | SSG (CLIP-RN50) | 46.38% | 47.78% | SSG (CLIP-ViT-B/16) | 63.58% | 65.20% |
>
> From the above table, we can find that our full model shows an obviously better performance than model (a). The reason is that geometric and photometric augmentation usually does not provide a more accurate prediction than the regular augmentation as shown in TPT/TDA, as they cause perturbations to the original image.
>
> **W2-(2)&Q2-(1):** *"Can you provide comparisons against simple baseline augmentations (e.g., random combinations of Gaussian noise and rotations) with PPD computation to demonstrate that your specific perturbation choices are essential?"*
>
> **A3**: Thanks for your suggestions. We supplement the comparison experiment between (a) PPD with Gaussian noise and rotations, and (b) PPD with shape patch-shuffle and style colour transformation with hue adjustment.
> | OOD Accuracy | (a)    | (b)    | OOD Accuracy | (a)    | (b)    |
> | ------------- | ------ | ------ | --------- | ------ | ------ |
> | SSG (CLIP-RN50)           | 46.02% | 47.78% | SSG (CLIP-ViT-B/16)       | 63.08% | 65.20% |
>
> From the above table, we can find that model (b) gives a much better performance than model (a). In fact, to obtain the PPD_st and PPD_sh, we need to corrupt/perturb the image's shape/style information, and patch-shuffle and colour transformation are suitable to reach this goal, because the augmented images obviously lose or change the shape/style information. But for the rotation and Gaussian noise augmentation, the shape information is perturbed slightly and the more than style information is destroyed, which even gives a negative impact on the performance.
>
> **Q2-(2)**: *"Additionally, please clarify whether the entropy computation uses the same shape and style perturbations as PPD, or different augmentations such as those employed in TPT?"*
>
> **A4**: Thanks for your question. We follow the augmentation in TPT/TDA to make the 63 augmented image views, and compute the entropy for these 64 (along with the raw image) images. For obtaining PPD_st and PPD_sh, we apply shape and style perturbations to the selected images (with high confidence) from 64 views, as described in Line182-Line183 and Line194-Line195.
>
> **W4&Q3**: *"Can you provide the missing ablation studies for ResNet-50, particularly showing the individual effects of PPD_sh and PPD_st, as well as the impact of patch shuffling on CNN-based architectures?"*
>
> **A5**: Thanks for your feedback. We supplement the ablation results with average OOD accuracy based on the CLIP-RN50 backbone.
>
> First, we supplement the ablation results about shape-sensitive and style-insensitive factors. Furthermore, we supplement the baseline method performance as described in **A8**. From the table, both shape-sensitive and style-insensitive factors give the better performance, while the combination gieves the best performance.
>
> | OOD Accuracy    | Baseline | PPD_st | PPD_sh | PPD    |
> | --------------- | -------- | ------ | ------ | ------ |
> | SSG (CLIP-RN50) | 46.33%   | 46.85% | 47.11% | 47.78% |
>
> Second, we report the ablation results about hyperparameters (patch number) on the shape perturbation (patch shuffle) below. From the table, the same conclusion is show as patch number as {4, 6} gives the better performance, and we set patch number as 4 for all experiments.
>
> | OOD Accuracy    | 2      | 4      | 6      | 8      | 16     |
> | --------------- | ------ | ------ | ------ | ------ | ------ |
> | SSG (CLIP-RN50) | 47.03% | 47.78% | 47.51% | 46.93% | 45.23% |
>
> Third, to show the effect of the patch-shuffle, we supplement the ablation experiments about the different shape perturbation types as shown below, corresponding to the results in Table 1 in the appendix. It is obvious that patch-shuffle gives the best performance. The reason is that, pixel-shuffle is difficult to clarify the background and object, center-occlusion misses the image where the object is not at center, and rotation does not impact the object shape information efficiently.
>
> | OOD Accuracy    | Pixel-shuffle | Center-occlusion | Patch-shuffle | Rotation |
> | --------------- | ------------- | ---------------- | ------------- | -------- |
> | SSG (CLIP-RN50) | 44.63%        | 46.79%           | 47.78%        | 47.03%   |
>
> **W5**: *"Typos."*
>
> **A6**: Thanks for your comments. We will correct all typos in the final version.
>
> **W6**: *"Target Distribution Access."*
>
> **A7**: Thanks for your clarification. We will modify the Line91 in Section 2 to "...yet they still require training on the target data with ground-truth labels. Differently, SSG aims to ..., where the model does not demand training on the target data with ground-truth labels.".
>
> **W7**: *"Method Foundation."*
>
> **A8**: Thanks for your feedback. We build our SSG on the naive cache-based method, and this baseline method only contains one positive cache and can be considered as the cache-based method TDA without a negative cache, which is different from the entropy-based TPT.
>
> For the cache implementation details of baseline and our SSG, we follow the TDA default setting for the positive cache. Specifically, the utilised cache is a dynamic key-value cache, whose memory size is 3 for all datasets following TDA. For the cache collection process, the key-value cache is initially empty and then accumulates a sufficient number of key-value pairs during the test-time adaptation.For the similarity function in adapted prediction, we utilise Eq.2 in the manuscript following TDA and Tip-Adapter.
>
> We will add the method foundation in the Subsection "Implementation details".

---

> > ### Comment · Reviewer_B6DJ · 2025-08-06
> >
> > I appreciate the authors' responses to all of my questions. Their explanations have adequately resolved my initial concerns.

---

> > > ### Author Response · Authors · 2025-08-06
> > >
> > > Dear Reviewer B6DJ,
> > >
> > > Thank you sincerely for the time you've dedicated to reviewing our work and the positive feedback on our rebuttal.
> > >
> > > We are grateful and encouraged that our response adequately addressed all of your concerns.
> > >
> > > Following your insightful suggestions, we guarantee that the key points discussed in the rebuttal will be incorporated into the final manuscript.
> > >
> > > Thank you again for your feedback and recognition. Your constructive comments have significantly contributed to enhancing the quality of our work.
> > >
> > > Best regards,
> > >
> > > Authors

---

### Official Review · Reviewer_8eZw · 2025-07-01

**Clarity:** 1
**Significance:** 3
**Originality:** 3
**Rating:** 5
**Confidence:** 5

**Summary:**

This paper proposes SSG, a training-free test-time adaptation method for CLIP models that adds shape and style awareness. It introduces Perturbed Prediction Difference (PPD) to measure prediction sensitivity to shape versus style changes. Using this, SSG reweights features to focus on shape-relevant signals and filters cache updates to retain only informative, uncertain samples. The method boosts zero-shot robustness under domain shift and outperforms prior work without extra tuning or high cost.

**Questions:**

1. How would you describe the SSG method, is it a cache with an adapter? I feel like the structure of SSG needs more introduction at the beginning of Section 3.2. If you are using a cache, what is the cache capacity? Can you please explain the cache collection process?

2. SSG updates the cache using both entropy and PPD criteria. How are these combined? For instance, do you require a sample to have both low entropy and high PPD to be cached, or is it a weighted combination? Can you also clarify the PPD criteria, does a higher PPD indicate better? This part is unclear. More details on the cache maintenance strategy would clarify the adaptation process.

3. The results on ImageNet are missing. Could you please provide results on the ImageNet dataset with a main comparison to TDA, DPE, and BoostAdapter for reference?

4. Could you please provide the results of the SSG+ method in Table 3 (cross-dataset generalization setting)?

5. Equation 4 lacks intuition. How can we see from it that high confidence can be harmful? Can you provide more evidence for this claim?

6. In the ablation studies, you introduce PPD reweighting and comparison. Could you identify all the operations under a single name, SSG or PPD, because it gets confusing what refers to what?

7. Can you show in Figure 3 the effect of reweighting and a comparison using the ViT backbone to make the ablation study more consistent?

8. There is no reference to Figure 1 in the manuscript. Can you please add the reference sentence in a future revision?

If the above concerns are addressed, I would be happy to raise my score. If space does not allow, feel free to skip Questions 6, 7, and 8.

**Ethical Concerns:**

["NO or VERY MINOR ethics concerns only"]

**Final Justification:**

As I mentioned earlier, I would increase my score if my concerns were addressed, therefore, I am raising my score to "Accept". The authors have now provided additional evidence supporting the applicability of their method. If the paper is accepted, I encourage the authors to open-source their work.

**Limitations:**

The current shape/style augmentation operations are basic and more sophisticated techniques could be used. The assumption that emphasizing shape always helps is not discussed as a potential weakness.

**Paper Formatting Concerns:**

The format is matched. The following typos are noted:
1) Line 342 - “generizable” -> “generalizable”
2) Line 258 - dot is missing
3) All occurrences of “adaption” should be changed to “adaptation”

**Quality:**

2

**Strengths And Weaknesses:**

**Strengths:**
- The motivation is well-defined: most existing methods rely on entropy-based criteria to select features for the cache.
- SSG is a novel approach that injects domain generalization insights into test-time adaptation of vision-language models.
- The method is training-free and computationally efficient.
- The results are strong. SSG achieves consistent accuracy gains over previous methods on two benchmarks. The enhanced version of the method, SSG+, further improves the performance.

**Weaknesses:**
- The method assumes shape features are always more reliable than style, but that’s not always true, some tasks depend heavily on style information. The paper doesn’t acknowledge this limitation.
- The comparison on the ImageNet dataset is missing, which weakens the evaluation. Additionally, the backbone used in the ablation study is inconsistent.
- The paper throws in too many abbreviations (SSG, SHS, SIS, PPD, STI), which makes it harder to follow and breaks the flow.
- Key details in the method description are underexplained. The Questions section below outlines several points that need clarification to improve the paper’s clarity.

---

> ### Author Rebuttal · Authors · 2025-07-31
>
> We sincerely thank you for your great efforts and insightful questions. We provide our feedback on weaknesses (W) and questions (Q) as follows.
>
> **W1**: *"The method assumes shape features are always more reliable than style, but that’s not always true."*
>
> **A1**: Thanks for your valuable feedback. Motivated by Shape-Bias[1] and MixStyle [2], SSG utilizes the shape features (shape-sensitive factors and style-insensitive factors) to represent the generalizable factors, aiming to improve model's robustness. Meanwhile, we agree that there are always some tasks (e.g., texture-based medical image analysis) that depend on style information, and we will supplement this limitation in the "Conclusion" section.
>
> [1] ImageNet-trained CNNs are biased towards texture; increasing shape bias improves accuracy and robustness, 2018ICLR; [2] Domain generalization with mixstyle, 2021ICLR.
>
> **W2-(1)&Q3**: *"The comparison result on the ImageNet dataset is missing."*
>
> **A2**: Thanks for your suggestion. We supplement the SSG performance on the ImageNet below.
> | CLIP-RN50    | ImageNet | CLIP-ViT-B/16 | ImageNet |
> | ------------ | -------- | ------------- | -------- |
> | TPT          | 60.74    | TPT           | 68.98    |
> | TDA          | 61.35    | TDA           | 69.51    |
> | DPE          | 63.41    | DPE           | 71.91    |
> | BoostAdapter | 61.54    | BoostAdapter  | 69.92    |
> | SSG          | 62.54    | SSG           | 71.21    |
> | SSG+         | 62.87    | SSG+          | 71.66    |
>
> As shown in the above table, SSG/SSG+ gives superior performance compared with training-free methods TDA/BoostAdapter.
>
> **W3&Q6**: *"Too many abbreviations (SSG, SHS, SIS, PPD, STI) in the paper."*
>
> **A3**: Thanks for your valuable feedback. For simplicity, we standardize the acronyms for "style-insensitive factors" as STI and "shape-sensitive factors" as SHS. Meanwhile, we maintain the SSG for the "style and shape guidance", and modify the perturbed prediction difference from PPD to SSG-score, while naming the operation "PPD reweight" as SSG-RW and "PPD comparison" as SSG-CP. We will modify all abbreviations in the final manuscript to make it clear.
>
> **Q1**: *"How would you describe the SSG method, is it a cache with an adapter?  What is the cache capacity? Can you please explain the cache collection process?"*
>
> **A4**: Thanks for your feedback. As you pointed out, our SSG can be described as a cache-based method with an adapter. As described in Section 3.1, SSG contains the dynamic visual cache, and SSG's final prediction is the combination of zero-shot prediction $P(f_{\mathrm{test}})$ from Equation 1 and the adapted prediction $P_{cache}(f_\mathrm{test})$ from Equation 2.
>
> We supplement more details about cache design to clarify. Following the positive cache in TDA, we utilize the dynamic key-value cache for SSG, and keep the cache capacity as 3 for all datasets. The key-value pair corresponds to the testing features $f_{\mathrm{test}}$ and the pseudo label $\hat{l}$ from the prediction $P(f_{\mathrm{test}})$.
>
> We supplement more details about cache collection for clarify. For the cache collection process, the key-value cache is initially empty and then accumulates a sufficient number of key-value pairs during the test-time adaptation. SSG progressively incorporates test predictions with lower update criterion while limiting cache capacity. Specifically, if the number of the collected key-value pair is less than the cache capacity (3 for SSG), current testing key-value will add to the cache directly. If the number of collected key-value pair is more than the capacity, SSG will choose the existing key-value pair with highest update criterion, and compare it with the current key-value pair, and add the key-value pair with lower update criterion to the cache.
>
> The update criterion is set to entropy of prediction $\mathrm{H}(P(f_{\mathrm{test}}))$ for previous cache-based method such as TDA or BoostAdapter, and our SSG modifies the update criterion as the combination of entropy and PPD (SSG-score), such as $ \mathrm{H}(P(f_{\mathrm{test}})) - \mathrm{PPD}$  as described in last paragraph (Line213 - Line218) in Section 3.2. Meanwhile, SSG only contain one cache without any other negative cache like TDA.
>
> Following your suggestion, we will add more introduction at the beginning of Section 3.2.
>
> **Q2**: *" How are entropy and PPD criteria combined? Does a higher PPD indicate better?"*
>
> **A5**: Thanks for your valuable question. For the first question, we combine the entropy and PPD criteria by the minus operation, such as $ \mathrm{H}(P(f_{\mathrm{test}})) - \lambda*\mathrm{PPD}$. Through this operation, a sample with low entropy and high PPD will be cached just as you suppose. We set $\lambda$ as 1 for all experiments, and the ablation experiments about the $\lambda$ are shown in Section B.2 in the appendix, which are shown below.
> | OOD Average | 0.5   | 1.0   | 2.0   | 5.0   |
> | ----------------------- | ----- | ----- | ----- | ----- |
> | SSG(CLIP-ViT-B/16)      | 64.91 | 65.20 | 64.73 | 64.59 |
>
> For the second question, a higher PPD means the sample contains more generalizable factors (shape-sensitive and style-insensitive factors), which is complementary to the entropy criteria for better performance. The ablation experiments about the effect of PPD reweighting (SSG-RW) and PPD comparison (SSG-CM) in Figure 3 in the manuscript show the improvement from the high PPD, and we show the results here.
> | OOD Accuracy        | PPD reweight | PPD comparison | Full Model |
> | ------------------- | ------------ | -------------- | ---------- |
> | SSG (CLIP-ViT-B/16) | 64.12        | 64.65          | 65.20      |
>
> **Q4**: *"Results of the SSG+ method in Table 3."*
>
> **A6**: Thanks for your question. We supplement the results of the SSG+ method in Table 3 below, and SSG+ shows the better performance in almost all datasets compared with SSG in Table 3 in the manuscript.
> | Method              | Aircraft | Caltech | Cars  | DTD   | EuroSAT | Flower | Food101 | Pets  | SUN397 | UCF101 | Ave.  |
> | ------------------- | -------- | ------- | ----- | ----- | ------- | ------ | ------- | ----- | ------ | ------ | ----- |
> | SSG+(CLIP-RN50)     | 22.14    | 90.68   | 60.99 | 52.31 | 44.58   | 69.07  | 78.91   | 86.32 | 64.41  | 64.77  | 63.42 |
> | SSG+(CLIP-ViT-B/16) | 31.97    | 95.24   | 69.97 | 55.47 | 62.54   | 74.64  | 86.99   | 91.89 | 70.64  | 72.69  | 71.20 |
>
> **Q5**: *"Equation 4 lacks intuition."*
>
> **A7**: Section A in the appendix shows more evidence for Equation 4, and we briefly summarise here.
>
> Equation 4 in the manuscript (Equation 10 in the appendix) is derived from the definition of the harmful samples. We define the harmful sample as one that reduces the differences in the mean logits between classes during the test-time procedure. Intuitively, we want the margin or differences between classes larger as much as possible, to obtain robust classification results.
>
> Based on this definition, we first use the properties of training-free (cache-based) VLM TTA to obtain the change in logits due to model's update in Equation 6,  shown as
>  $\Delta\left(w\cdot\mathbf{v}\left(\mathbf{x}^{\text {test}}\right)\right)=\Delta w(\mathbf{x})\cdot\mathbf{v}\left(\mathbf{x}^{\text {test}}\right) \approx \mathbf{v}(\mathbf{x}) \cdot \mathbf{v}\left(\mathbf{x}^{\text {test}}\right)$,
> where stored visual feature can be regarded as the residual update of the raw classifier.
> Then we obtain the change in the gap between the mean logits of samples from two classes as Equation 7, shown as $\mathbf{v}(\mathbf{x})\cdot(E_{X_{+1}^{{test}}}[\mathbf{v}(\mathbf{x}^{{test}})]-E_{X_{-1}^{{test}}}[\mathbf{v}(\mathbf{x}^{{test}})]).$
> Finally, based on the disentangled factors, we approximate Equation 7 to Equation 10 (Equation 4 in the manuscript) as $\mathbf{v_pp}(\mathbf{x})\cdot(E_{X_{+1}^{ {test }}}[\mathbf{v_pp}(\mathbf{x}^{ {test }})]-E_{X_{-1}^{{test}}}[\mathbf{v_pp}(\mathbf{x}^{{test}})])+\mathbf{v_pn}(\mathbf{x})\cdot(E_{X_{+1}^{ {test}}}[\mathbf{v_pn}(\mathbf{x}^{{test}})]-E_{X_{-1}^{{test}}}[\mathbf{v_pn}(\mathbf{x}^{{test}})]).$
>
> To make the differences large, we need to make Equation 10 positive. According to Equation 9, the term multiplied by v_pp is positive, while the term multiplied by v_pn is negative, so we need to highlight the generalizable factors v_pp. Meanwhile, according to Equation 10, the sample with high confidence but with significant v_pp factors is also harmful for the differences between the two classes.
>
> **Q7**: *"Can you show in Figure 3 the effect of reweighting and a comparison using the ViT backbone?"*
>
> **A8**: We supplement the average OOD accuracy results about the effect of reweighting (SSG-RW) and comparison (SSG-CM) based on the ViT-B/16 backbone below. According to the results, we can find that both reweighting and comparison operations give an improvement,  and a combination of them gets the best performance.
> | OOD Accuracy        | PPD reweight | PPD comparison | Full Model |
> | ------------------- | ------------ | -------------- | ---------- |
> | SSG (CLIP-ViT-B/16) | 64.12        | 64.65          | 65.20      |
>
> Moreover, we give the OOD average accuracy for the ablation results in the middle of Figure 3 to make the ablation consistent. This ablation experiment is about the patch number on the shape perturbation (patch shuffle). According to results, we set the patch number to 4 for all experiments.
> | OOD Accuracy        | 2      | 4      | 6      | 8      | 16     |
> | ------------------- | ------ | ------ | ------ | ------ | ------ |
> | SSG (CLIP-ViT-B/16) | 64.38% | 65.20% | 64.91% | 63.47% | 62.52% |
>
> **Q8&Typos**: *"There is no reference to Figure 1 in the manuscript. And Formatting Concerns."*
>
> **A9**: Thanks for your valuable suggestions. We refer the Figure 1 in Line203, and we will move it to the first paragraph of Section 3.2 in the final version. And we will correct all typos in the final version.

---

### Official Review · Reviewer_BH2W · 2025-07-02

**Clarity:** 2
**Significance:** 3
**Originality:** 3
**Rating:** 4
**Confidence:** 4

**Summary:**

This paper proposes a training-free test-time adaptation method for vision-language models, called Shape and Style Guidance (SSG). SSG aims to improve robustness to domain shifts by incorporating generalizable visual factors, specifically shape-sensitive and style-insensitive cues. It perturbs test-time images using shape and style transformations and measures the prediction difference compared to the original, forming a Perturbed Prediction Difference (PPD). This signal is used to reweight high-confidence visual features and guide cache updates, emphasizing features that are robust and informative. Experiments on several out-of-distribution and cross-domain benchmarks show that SSG achieves strong improvements over existing methods.

**Questions:**

1. Could you please clarify whether SSG is implemented on top of TDA, Tip-Adapter, or another cache-based method? It would also be helpful to specify the cache structure (e.g., queue types and sizes, similarity function), and explain whether the entropy-minus-PPD criterion applies to both the positive and negative caches. If so, how are negative PPD values handled?

2. Could you please include or comment on an augmentation-only ablation—e.g., applying shape and style perturbations in TDA without PPD-based reweighting or cache updates?

3. How is the parameter k* determined? Is it a fixed value, dataset-dependent, or adaptively selected per image? Please consider clarifying this in the main text.

4. When computing PPD, are the same top-k* low-entropy indices used across the original, shape-perturbed, and style-perturbed augmentations? If so, please make this explicit, as it’s important for reproducibility.

5. SSG appears to use roughly 3× more augmentations than TDA, yet reports similar inference time. Could you please clarify whether forward passes are reused or shared across branches?

6. Have you tested SSG on other vision-language models, such as SigLIP or SigLIP-2? It would be helpful to know if the approach generalizes beyond CLIP.

7. In Table 1, the “ensemble” row is somewhat ambiguous. Could you please explain whether this refers to CLIP predictions averaged over all augmentations, or something else?

**Ethical Concerns:**

["NO or VERY MINOR ethics concerns only"]

**Final Justification:**

I thank the authors for their thorough and well-organized rebuttal. The additional clarifications, ablation studies, and efficiency/memory analyses have satisfactorily addressed most of my concerns.

While I feel the methodological novelty is moderate, the integration of shape-sensitive and style-insensitive factors offers an intresting enhancement.

Given that my other concerns have been addressed, I am raising my score to 4.

**Limitations:**

yes

**Paper Formatting Concerns:**

No Formatting Concerns

**Quality:**

3

**Strengths And Weaknesses:**

**Strengths**

1. **Well Written.** The paper is well written and clearly structured, making the proposed method easy to follow and understand.

2. **Interesting Idea.** Introducing shape-sensitive and style-insensitive cues for test-time adaptation is a novel and compelling idea that adds a new perspective to training-free adaptation.

3. **Practicality.** The method builds on training-free adaptation techniques and avoids gradient-based updates, making it more practical and efficient for real-world deployment compared to optimization-based approaches.

4. **Thorough empirical validation.** The experimental evaluation is thorough, with comparisons against a wide range of state-of-the-art baselines across both out-of-distribution and cross-domain benchmarks, where the proposed method consistently outperforms others.



**Weakness**

1. **Novelty.** While the paper is thoughtfully designed and thoroughly evaluated, the core idea of perturbation‐based reweighting feels like incremental improvement of existing cache-based TTA techniques rather than a transformative advance.

2. **Efficiency concerns.** SSG is reported to have similar inference time to TDA, despite using roughly three times more augmentations per image (original, shape-perturbed, and style-perturbed). Unless forward passes are reused, this seems inconsistent. Clarifying these details is important to accurately assess the true computational cost.

3. **Cache design ambiguity.** The paper does not clearly specify whether SSG is built on TDA, Tip-Adapter, or another cache-based method, leaving key implementation details—such as queue structure, and memory size—undefined. Additionally, if based on TDA, it is not explained how the entropy-minus-PPD criterion affects the negative cache or whether it is applied symmetrically. These ambiguities limit interpretability and reproducibility.

4. **Missing augmentation-only ablation.** A missing ablation is a version of TDA that uses the same 3× augmentation strategy as SSG (original + shape + style), but without applying the PPD-based reweighting or entropy-PPD cache update. This would isolate the benefit of the diverse views themselves versus the actual contribution of SSG's mechanisms.

5. **Missing details and analysis of** k*. One parameter,  k* , determines how many low-entropy augmentations are used for prediction reweighting and cache updates. However, the paper does not specify how k* is selected, nor does it study its impact on performance. Given its central role in PPD computation and feature aggregation, a sensitivity analysis or ablation would help assess the method's robustness.

6. **Inconsistent ablations.** While the ablation studies are informative, they are inconsistent. Different components are analyzed on different datasets (e.g., ImageNet-V2, -R, -A, -S) and with different backbones (ResNet vs. ViT), making it difficult to compare relative contributions. A more rigorous evaluation would unify ablations on a consistent benchmark or report average OOD accuracy for all variants.

7. **Missing statistical variance in main results.** Although the appendix includes significance testing, no variance metrics (e.g., standard deviations) are reported in the main tables. Given the sensitivity of test-time adaptation to augmentation noise and pseudo-labeling, reporting mean ± std across runs would improve transparency and trust in the reported gains.


8. **Minor Issues.**
- The paper likely uses the same top-k* indices across original and perturbed augmentations when computing PPD, but this should be stated explicitly to avoid confusion.
- The subsection “Hyperparameters about Style-Insensitive Factors” focuses on the weighting between PPD_sh and PPD_st, but omits the actual style perturbation settings. It may be helpful to rename or expand the section to include these details.
- Figure and terminology consistency. In the caption of Figure 1, both branches are labeled PPD_sh, though the right should be PPD_st. Acronyms for style-insensitive factors (e.g., “SIS” vs. “STI”) should also be standardized for clarity.

---
I welcome the authors' responses to the above concerns and am open to revising my assessment accordingly based on their clarifications.

---

> ### Author Rebuttal · Authors · 2025-07-31
>
> We appreciate your thorough review of our paper, along with your constructive comments. We provide our feedback on weaknesses (W) and questions (Q) as follows.
>
> **W1**: *"While the paper is thoughtfully designed and thoroughly evaluated, the core idea feels like an incremental improvement of existing cache-based TTA techniques."*
>
> **A1**: Thank you for the approval of our thoughtful design and thorough evaluation. While our SSG is easy to implement with existing cache-based VLM TTA methods, we do not think it is an incremental improvement.
>
> First, SSG reveals the effect of generalizable factors for the cache-based (training-free) VLM TTA, which has not been studied before. Similar to TDA (2024CVPR) or BoostAdapter (2024NeurIPS), our SSG gives a new perspective for the cache-based TTA framework, and the corresponding experimental results prove the effectiveness.
>
> Second, we give the theoretical analysis for the generalizable factors under training-free TTA scenarios. Different from the entropy minimization scenarios, it is not trivial to derive the proof based on the properties of cache-based methods, as shown in Line15 - Line21 in the Appendix.
>
> Third, SSG gives a potential direction to explore a further cache-based approach by designing more advanced generalizable factors.
>
> **W2&Q5**: *"Efficiency concern. Could you please clarify whether forward passes are reused or shared across branches?"*
>
> **A2**:  Thanks for your thoughtful feedback. The forward passes are indeed shared across branches in our SSG, and we give the detailed explanations below.
>
> First, SSG utilised shape and style augmentation is extremely simple to implement. For the shape augmentation, we use the patch-shuffle operation, which is efficiently implemented by `erarrange func` in `einops packages`. For the style augmentation, we use the colour transformation with hue adjustment, which is easily implemented by the `kornia package`.
>
> Second, SSG applies the shape/style augmentation to the raw input images before the model forward procedure, and we concatenate them into a batch to feed into the model, thus the forward passes can be shared and maintain the forward time nearly unchanged.
>
> Third, after model forward and obtaining the logits, the following PPD calculation, reweighting and cache update are simple mathematical calculations, which do not introduce much more inference burden.
>
> Finally, we give the detailed inference time comparison below. SSG needs a little more data processing time due to the shape/style augmentation, and the total increased inference time is relatively small.
> | Methods    | Data Process | Model Forward | Rest Part | Total  |
> | ---------- | ------------ | ------------- | --------- | ------ |
> | TDA        | 15 ms        | 82 ms         | 34 ms     | 132 ms |
> | SSG (Ours) | 21 ms        | 83 ms         | 35 ms     | 139 ms |
>
> Furthermore, we supplement the efficiency comparison in ImageNet-V, which contains 10000 images, in the following table.
> | Methods    | Testing Time | OOD Accuracy |
> | ---------- | ------------ | ------------ |
> | TDA        | 22 min       | 63.89        |
> | DPE        | 41 min  | 64.43        |
> | SSG (Ours) | 23 min 10 s       | 65.20        |
>
> From the above table, even for the 1w image scale, our SSG only increase 1 minute compared with TDA while improving the OOD accuracy by about 1.3%. Our SSG only demands 56% inference time compared with DEP while achieving better performance.
>
> **W3&Q1**: *"Cache design ambiguity."*
>
> **A3**: Thank you for pointing it out. We build our SSG on the naive cache-based method, which only contains the positive cache and can be considered as the TDA without a negative cache.
>
> For the cache implementation details, we follow the TDA default setting for the positive cache. Specifically, the utilised cache in SSG is a dynamic key-value cache, whose memory size is 3 for all datasets. For the similarity function in adapted prediction, we utilise Eq.2 in the manuscript following TDA and Tip-Adapter.
>
> Last but not least, we combine BoostAdapter with SSG as SSG+. In SSG+, we only apply our SSG operations for the positive cache in BoostAdapter, while keeping the negative cache and related operations identical to BoostAdapter.
>
> **W4&Q2**: *"Missing augmentation-only ablation."*
>
> **A4**: Thanks for your suggestions. We supplement the augmentation-only ablation results below. Model-(a) means our SSG applying shape and style perturbation without PPD-based reweighting or cache updates, and model-(b) means our full SSG.
> | OOD Accuracy        | (a)    | (b)    |
> | ------------------- | ------ | ------ |
> | SSG (CLIP-RN50)     | 46.38 | 47.78 |
> | SSG (CLIP-ViT-B/16) | 63.58 | 65.20 |
>
> From the above table, we can find that our full model shows an obviously better performance than model (a).
>
> **W5&Q3**: *"Details and analysis of k\*."*
>
> **A5**: Thanks for your questions. We follow the operation in TPT and TDA, choosing k* as the top 10%,  and we keep the k* value for all experiments. We supplement the ablation results about the k* value, as shown below.
> | OOD Accuracy        | 5%     | 10%    | 15%    | 20%    |
> | ------------------- | ------ | ------ | ------ | ------ |
> | SSG (CLIP-RN50)     | 47.45 | 47.78 | 47.59 | 47.55 |
> | SSG (CLIP-ViT-B/16) | 64.89 | 65.20 | 65.04 | 65.02 |
>
> As shown in the above table, we can find that setting k* as the top-10% achieves the best performance, which is consistent with the conclusion from TPT.
>
> **W6**: *"Inconsistent ablations."*
>
> **A6**: Thanks for your valuable suggestions. We unify the ablation experiment based on the ViT-B/16 backbone and report the average OOD accuracy.
>
> First, we report the ablation results about shape-sensitive and style-insensitive factors. As shown in Fig.3 in the manuscript, we have given the comparison results, and we summarise below, while we supplement the baseline performance on which we built our SSG as stated in **A3**.
> | OOD Accuracy        | Baseline | PPD_st | PPD_sh | PPD    |
> | ------------------- | -------- | ------ | ------ | ------ |
> | SSG (CLIP-ViT-B/16) | 63.46   | 64.08 | 64.54 | 65.20 |
>
> Second, we report the ablation results about hyperparameters (patch number) on the shape perturbation (patch shuffle) below. From the table, the same conclusion is shown as patch number as {4, 6} gives the better performance, and we set patch number to 4 for all experiments.
> | OOD Accuracy        | 2      | 4      | 6      | 8      | 16     |
> | ------------------- | ------ | ------ | ------ | ------ | ------ |
> | SSG (CLIP-ViT-B/16) | 64.38 | 65.20 | 64.91 | 63.47 | 62.52 |
>
> Third, the ablation results about the combination hyperparameters between PPD_st and PPD_sh are shown in Line317 in the manuscript, and we omit them for saving space.
>
> Fourth, we report the ablation results about the effect of the PPD operation below. According to the results, we can find that both reweighting and comparison operations give an improvement,  and a combination of them gets the best performance.
> | OOD Accuracy        | PPD reweight | PPD comparison | Full Model |
> | ------------------- | ------------ | -------------- | ---------- |
> | SSG (CLIP-ViT-B/16) | 64.12       | 64.65         | 65.20     |
>
> **W7**: *"Missing statistical variance in main results. Reporting mean ± std across runs would improve transparency and trust in the reported gains."*
>
> **A7**: Thanks for your feedback. We supplement the statistical variance and show it below. Due to the space limitation, more variance will be provided in the final manuscript.
> | Method               | ImageNet-A   | ImageNet-V   | ImageNet-R   | ImageNet-S   | OOD Average  |
> | -------------------- | ------------ | ------------ | ------------ | ------------ | ------------ |
> | SSG (CLIP-RN50)      | 31.54 (0.25) | 56.78 (0.18) | 63.77 (0.20) | 39.11 (0.12) | 47.78 (0.19) |
> | SSG (CLIP-ViT-B/16)  | 62.02 (0.14) | 65.32 (0.12) | 81.33 (0.18) | 52.13 (0.13) | 65.20 (0.13) |
>
> **W8-(1)&Q4**:  *"When computing PPD, are the same top-k* low-entropy indices used across the original, shape-perturbed, and style-perturbed augmentations?"*
>
> **A8**: We indeed utilise the same top-k* low-entropy indices used across the original, shape-perturbed and style-perturbed augmentations.
>
> **W8-(2)**: *"Rename the subsection."*
>
> **A9**: We will rename this subsection as "Hyperparameters about the combination between PPD_st and PPD_sh" to clarify.
>
> **W8-(3)**: *"Typos."*
>
> **A10**: We will accordingly modify the typos and standardise the acronyms for style-insensitive factors as STI.
>
> **Q6**:  *"Have you tested SSG on other vision-language models?"*
>
> **A11**: Thanks for your constructive comments. Our SSG can easily be applied to other vision-language models, and we follow DPE to involve the OpenCLIP (ViT-L/14) as an example for a fair and public comparison. We show the comparison results below.
> |                     | ImageNet-A | ImageNet-V | ImageNet-R | ImageNet-S | OOD Average |
> | ------------------- | ---------- | ---------- | ---------- | ---------- | ----------- |
> | OpenCLIP (ViT-L/14) | 53.88      | 67.69      | 87.42      | 63.18      | 68.13       |
> | TDA                 | 61.27      | 68.42      | 88.41      | 64.67      | 70.69       |
> | DPE                 | 61.09      | **70.83**  | 89.18      | 66.33      | 71.86       |
> | SSG (Ours)          | **62.85**  | 69.97      | **89.67**  | **66.56**  | **72.26**   |
>
> We can observe that our SSG still outperforms training-free  TDA by 1.57%/ on average across 4 datasets, showing that our method generalizes well to larger-scale VLMs. And more vision-language models such as SigLIP or DFN will be explored in the future.
>
> **Q7**: *"The ensemble row is somewhat ambiguous."*
>
> **A12**: We follow TPT and DPE to involve the "Ensemble" as the comparison method.  "Ensemble" means the baseline zero-shot CLIP prediction using the ensemble of 80 hand-crafted prompts from [1].
>
> [1] Learning transferable visual models from natural language supervision. ICML 2021.

---

### Note · Authors · 2025-08-13

We sincerely thank the AC and all reviewers for their time, effort, and constructive feedback.

We have addressed ALL concerns from R(eviewer)-BH2W, 8eZw and B6DJ. R-BH2W states "additional clarifications ... analyses have **satisfactorily addressed my comments**.", R-8eZw states "**all my main concerns** **were addressed**", and R-B6DJ states "explanations have **adequately resolved my concerns**.". Meanwhile, we have "addressed most of the initial concerns" from R(eviewer)-rBMJ, and gave detailed responses to show our SSG's promising innovations in the rebuttal/discussion period, with no further discussions proposed till the end.

We are happy to have addressed concerns from Reviewers, and all provided experiments and clarifications have verified the effectiveness of our SSG comprehensively.

Moreover, we believe our SSG shows promising innovation compared with existing works including DeYO.

We thank R-BH2W for the support about "shape-sensitive and style-insensitive factors offer an interesting enhancement " and "adds a new perspective to training-free adaptation", and thank R-8eZW for "SSG is a novel approach that injects DG insights into TTA of VLM". Moreover, we appreciate our clarification about the difference from DeYO' has solved concerns from R-B6DJ, and R-rBMJ thinks our work gives a "valuable contribution".

Our SSG introduces shape-sensitive and style-insensitive factors for efficient (training-free) VLM TTA for the first time, which is different from the (VLM) entropy-optimization-bp TTA setting and framework. Second, our SSG derives the theoretical analysis from the property of cache-based methods in a non-trivial way, while designing suitable and efficient operations. Then, our SSG verifies generalizable factors on VLM models with strong performance and efficiency, as DeYO and other works have not explored before. Finally, our SSG considers the style-related factors, which have been ignored before, through detailed experiments.

We believe our SSG gives a new perspective to efficient (training-free) VLM TTA, with practical scenarios, reasonable theoretical analysis, efficient operations and strong performance. Meanwhile, we fully respect the insight and contribution from previous methods including DeYO, and we will give more discussions to make it more explicitly credited in the final version.

Once again, we thank all reviewers for their valuable suggestions, and we will incorporate the content of the rebuttal into the final version.

---

### Decision · Program_Chairs · 2025-09-17

**Decision:**

Accept (poster)

**Comment:**

This paper proposes SSG, a training-free test-time adaptation method for CLIP models that adds shape and style awareness. It introduces Perturbed Prediction Difference (PPD) to measure prediction sensitivity to shape versus style changes. Using this, SSG reweights features to focus on shape-relevant signals and filters cache updates to retain only informative, uncertain samples.  The method demonstrates improvements over existing approaches across two benchmarks using different backbone architectures.

The paper initially received an average rating of 3 (borderline reject). T During rebuttal, the authors answered the reviewers questions to the satisfaction of the reviewers and two reviewers increased scores to "accept" and the other two to "borderline accept". There is some concern that the novelty of the method is incremental over the previous DeYO approach.

The ACs feel that the paper clearly meets the standards of a NeurIPS paper. The authors are encouraged to include some of the explanations and new results presented in the rebut in their revised paper.